# EconGym: A Scalable AI Testbed with Diverse Economic Tasks

**Qirui Mi**[1,2,3]**, Qipeng Yang**[4,5,7]**, Zijun Fan**[4,5,7]**, Wentian Fan**[4,5,7]**, Heyang Ma**[1,5,7]**,
Chengdong Ma**[6]**, Siyu Xia**[1,2]**, Bo An**[3]**, Jun Wang**[8]**, Haifeng Zhang**[1,2,7]*

[1]Institute of Automation, Chinese Academy of Sciences
[2]School of Artificial Intelligence, Chinese Academy of Sciences
[3]Nanyang Technological University    [4]Nanjing University of Posts and Telecommunications
[5]University of Chinese Academy of Sciences, Nanjing    [6]Peking University
[7]Nanjing Artificial Intelligence Research of IA    [8]University College London

## Abstract

Artificial intelligence (AI) has become a powerful tool for economic research, enabling large-scale simulation and policy optimization. However, applying AI effectively requires simulation platforms for scalable training and evaluation—yet existing environments remain limited to simplified, narrowly scoped tasks, falling short of capturing complex economic challenges such as demographic shifts, multi-government coordination, and large-scale agent interactions. To address this gap, we introduce **EconGym**, a scalable and modular testbed that connects diverse economic tasks with AI algorithms. Grounded in rigorous economic modeling, EconGym implements 11 heterogeneous role types (e.g., households, firms, banks, governments), their interaction mechanisms, and agent models with well-defined observations, actions, and rewards. Users can flexibly compose economic roles with diverse agent algorithms to simulate rich multi-agent trajectories across **25+ economic tasks** for AI-driven policy learning and analysis. Experiments show that EconGym supports diverse and cross-domain tasks—such as coordinating fiscal, pension, and monetary policies—and enables benchmarking across AI, economic methods, and hybrids. Results indicate that richer task composition and algorithm diversity expand the policy space, while AI agents guided by classical economic methods perform best in complex settings. EconGym also scales to **100k** agents with high realism and efficiency.

## 1 Introduction

Artificial intelligence (AI) has emerged as a powerful computational tool for addressing complex economic problems characterized by high-dimensional data, dynamic evolution, and large-scale heterogeneous agent interactions [4, 30]. Recent advances have demonstrated AI's promise in solving previously intractable problems in complex economic environments [48]. For example, DeepHAM [15] leverages deep learning to approximate solutions for high-dimensional Heterogeneous Agent Models (HAMs); the AI Economist [47] pioneered reinforcement learning (RL) [41] in optimal taxation; TaxAI [27] further demonstrated the effectiveness of multi-agent RL in complex dynamic tax environments; and Dynamic-SMFG [29] proposed data-driven MARL methods to solve asymmetric games between governments and agent populations. More recently, large language models (LLMs) have shown promise in capturing human-like decision behavior in both micro and macroeconomic contexts [20], as seen in EconAgent [23] and AgentSociety [34].

---

*Corresponding author: `haifeng.zhang@ia.ac.cn`

39th Conference on Neural Information Processing Systems (NeurIPS 2025) Track on Datasets and Benchmarks.

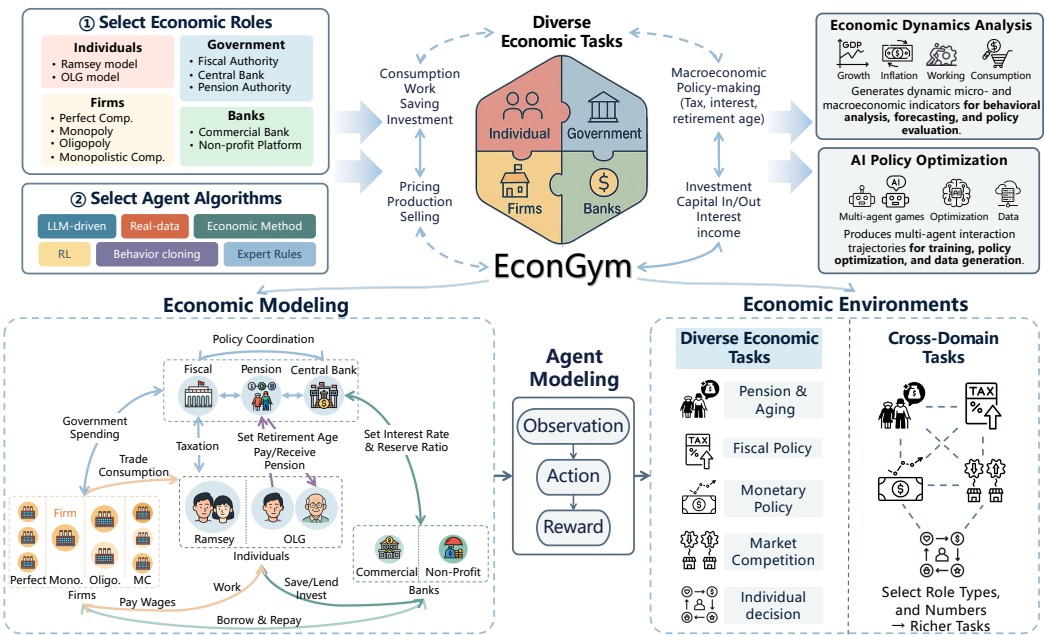

Figure 1: Overview of **EconGym**. Users define tasks by selecting economic roles and agent algorithms, generating dynamic multi-agent trajectories. These trajectories support economic analysis for the economics community and policy optimization for AI community. Built on rigorous economic theory and modular agent modeling, EconGym enables diverse and cross-domain economic tasks.

Despite this progress, realizing the full potential of AI in economics relies critically on simulation platforms for training, evaluation, and benchmarking [36]. Yet current platforms suffer from three key limitations (Table 1): **(1) Most environments are tailored to simplified scenarios.** For instance, the AI Economist [47] models a taxation scenario where agents collect wood and stone—far from real-world complexity—leading to poor transferability of learned policies. **(2) The scope of economic tasks is highly restricted.** Platforms like EconoJax [35], TaxAI [27], and EconAgent [23] focus solely on taxation; R-MABM [6] addresses market competition; ABIDES-Economist [12] supports only labor and market scenarios. Many pressing economic questions thus remain out of reach for AI exploration. **(3) Existing environments typically isolate a single economic issue while holding others fixed.** This limits the exploration of cross-domain tasks, such as how taxation, monetary policy, and labor markets jointly evolve. Policies designed in isolation often fail to generalize in more complex, integrated settings [43].

To address these challenges, we introduce **EconGym**—a modular, theory-grounded, and extensible platform for training and evaluating AI algorithms across diverse and interconnected economic problems (Fig. 1). EconGym provides the following key contributions:

- **Rigorous and extensible economic modeling.** EconGym models both micro- and macroeconomic dynamics using advanced economic theory [25], with explicit interactions among four core roles—households, firms, banks, and governments. Each role includes multiple heterogeneous agent types (e.g., OLG households, consumers, monopolists, central banks), providing a unified theoretical foundation for simulating diverse economic problems.

- **Composable environments for diverse and cross-domain economic tasks.** Built on this foundation, EconGym models all heterogeneous economic types as modular agents with well-defined observations, actions, and rewards. This modularity enables flexible composition of agent roles and numbers to construct a wide range of economic tasks. Examples include pension reform (OLG + pension authority), tax optimization (Infinitely-lived agent + fiscal authority), monetary transmission (central + commercial banks), and cross-domain policy coordination (fiscal + monetary + firm + household agents). While this paper presents 25 example tasks, the platform supports far more through compositional generalization.

- **A unified testbed for AI algorithm development.** EconGym supports multiple agent algorithms and their combinations, including reinforcement learning (RL), large language models (LLMs),

behavior cloning (BC), rule-based strategies, economic solvers, and real-data-driven agents—and scales to populations of up to 100,000 agents. This provides a scalable environment for training, benchmarking, and improving AI-based policies.

Using EconGym is simple: users define a task by selecting agent roles and assigning algorithms, then simulate agent–environment interactions to generate dynamic behavioral trajectories (Fig. 1). For economists, EconGym enables reproducible policy analysis and behavioral evaluation. For AI researchers, it provides a structured multi-agent testbed for algorithm development and benchmarking.

We validate EconGym's capabilities through comprehensive experiments: **(1) AI policy learning.** In the aging-pension task, RL excels in optimizing long-term sustainability as a social planner, while LLMs better capture human-centric objectives (Fig. 4). **(2) Benchmark testbed.** EconGym demonstrates that richer task composition and algorithm diversity expand the policy space. In cross-domain settings, coordinated fiscal, pension, and central bank policies reveal synergies and conflicts (Fig. 5, Table 5). **(3) Scalability and realism.** The platform simulates up to **100k** agents while maintaining both accuracy and efficiency (Fig. 6, Fig. 7). By bridging AI and economics, EconGym offers a scalable testbed for diverse economic decision-making tasks. Code is available at `https://github.com/Miracle1207/EconGym`.

Table 1: Comparison of simulation platforms for AI-based economic decision-making.

| Platform | Tasks Diversity | Cross-domain | AI Agent Support | Agent Scale (Individuals) | Real-Data Calibration | Heterogeneous Role Types |
|---|---|---|---|---|---|---|
| AI Economist [47] | 1 (Tax) | ✗ | RL | 10 | ✗ | 2 |
| EconoJax [35] | 1 (Tax) | ✗ | RL | 100 | ✗ | 2 |
| R-MABM [6] | 1 (Market) | ✗ | RL | 1k | ✗ | 4 |
| ABIDES-Econ. [12] | 2 (Labor, Market) | ✗ | RL | 2 | ✓ | 4 |
| TaxAI [27] | 1 (Tax) | ✗ | RL | 10k | ✓ | 4 |
| EconAgent [23] | 1 (Tax) | ✗ | LLM | 100 | ✗ | 3 |
| AgentSociety [34] | 1 (Basic Income) | ✗ | LLM | >10k | ✗ | 4 |
| **EconGym (Ours)** | >25 (Pension, Tax, Money, Market, etc.) | ✓ (e.g. Multi-Government Coordination) | LLM, RL | 100k | ✓ | 11 |

## 2 Related Work

**AI for Economic Decision-Making** Artificial intelligence, particularly reinforcement learning (RL) and large language models (LLMs), have shown strong promise in economic decision-making. The AI Economist [47] applies PPO to optimize tax schedules, and TaxAI [27] leverages multi-agent RL for tax policies. RL has also been applied to fiscal crisis management [47], monetary policy [19, 8], international trade [39], externality pricing [9], household savings and consumption [40, 37, 3], general equilibrium [22, 18], and barter behaviors [21, 32]. DSMFG [29] introduces a data-driven framework for asymmetric interactions between governments and large populations.

LLM-driven agents have gained traction in economic simulation. Homo Silicus agents [20] simulate behavior in economic experiments, while Generative Agents [33] exhibit emergent dynamics in social simulations. EconAgent [23] applies LLMs to tax modeling, and other studies explore LLMs in public policy [16], consumption [11], and trading [45, 46]. Mean-field LLMs [28] support population dynamics simulation, and hybrid methods [7] combine LLMs with RL to improve decision-making. Despite these advances, robust simulation platforms are essential for AI training and evaluation.

**Economic Simulation Platforms for AI Research** The intersection of AI and economics has given rise to simulation platforms designed for agent training and algorithmic evaluation. The AI Economist [47] pioneered tax policy learning but features limited agent interaction. EconoJax [35] accelerates tax simulations using JAX, while R-MABM [6] explores macroeconomic rationality with 1,000 RL agents. However, these environments are restricted to narrow domains, making it difficult for learned policies to generalize. ABIDES-Economist [12] incorporates real-world data to simulate household–firm dynamics at a micro scale. TaxAI [27] applies multi-agent RL across governments, firms, and households. EconAgent [23] uses LLMs for macroeconomic reasoning, though its scalability is limited. AgentSociety [34] enables large-scale LLM simulations but lacks economic structure and agent role flexibility. While each platform makes valuable contributions, most remain limited in task diversity, algorithmic flexibility, or agent heterogeneity. To address these challenges, we introduce **EconGym**—a modular and scalable testbed supporting diverse economic tasks for rigorous AI evaluation.

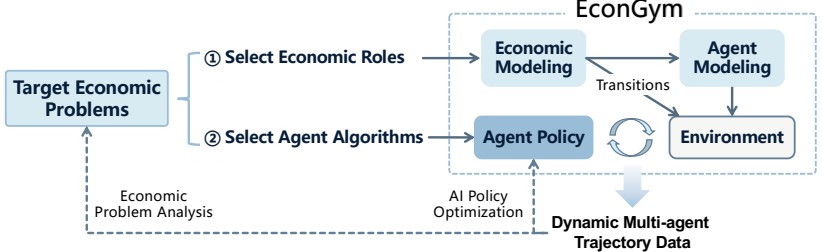

Figure 2: Workflow of EconGym.

# 3 The Structure of EconGym

This section first introduces the workflow of EconGym, followed by its core components: economic modeling (§3.1) and agent modeling (§3.2).

**Workflow of EconGym.** To solve a target economic problem, users follow two steps (Fig. 2):

**(1) Select economic roles relevant to the problem**, as illustrated in Table 2. Available roles are detailed in Section 3.1. Based on this selection, EconGym constructs the underlying economic environment by instantiating all relevant roles, their interactions, and environment transitions (§3.1). These are then automatically transformed into modular agents (§3.2)—each with a defined observation space, action space, and reward—forming a Markov game environment.

**(2) Select agent algorithms to drive on economic roles.** The selected algorithms drive the policies of economic agents and interact with the environment to generate dynamic multi-agent trajectories.

For AI research, these trajectories support policy optimization, algorithmic innovation, and evaluation. From an economics perspective, they contain rich indicators computed by EconGym, enabling the analysis of complex economic phenomena.

Table 2: Economic problems and recommended economic role types in EconGym.

| Economic Problems | Government Type | Individual Type | Firm Type | Bank Type |
|---|---|---|---|---|
| **Pension Issues** | | | | |
| Q1: How does delayed retirement affect the economy? | Pension | OLG | Perfect | Non-profit |
| Q2: Do personal pensions improve security? | Pension | OLG | Perfect | Non-profit |
| Q3: How does aging impact the macroeconomy? | Pension | OLG | Perfect | Non-profit |
| Q4: How to close pension funding gaps? | Pension | OLG | Perfect | Non-profit |
| Q5: How do pension systems vary across countries? | Pension | OLG | Perfect | Non-profit |
| **Fiscal Policy Issues** | | | | |
| Q6: Can consumption taxes boost growth and fairness? | Fiscal | Ramsey | Perfect | Non-profit |
| Q7: How does inheritance tax affect wealth distribution? | Fiscal | OLG | Perfect | Non-profit |
| Q8: Does universal basic income enhance equity? | Fiscal | OLG | Perfect | Non-profit |
| Q9: How to design optimal tax policies? | Fiscal | Ramsey | Perfect | Non-profit |
| Q10: How does wealth tax impact wealth concentration? | Fiscal | Ramsey | Perfect | Non-profit |
| **Monetary Policy Issues** | | | | |
| Q11: How effective are negative interest rates? | Central Bank | Ramsey | Perfect | Commercial |
| Q12: How to control inflation via monetary policy? | Central Bank | Ramsey | Perfect | Commercial |
| Q13: What are the long-term effects of quantitative easing? | Central Bank | Ramsey | Perfect | Commercial |
| Q14: How to set optimal bank rate spreads? | Central Bank | Ramsey | Perfect | Commercial |
| Q15: How to coordinate monetary and fiscal policies? | Central Bank + Fiscal | Ramsey | Perfect | Commercial |
| **Market Competition Issues** | | | | |
| Q16: How does technology drive long-term growth? | Fiscal | OLG | Perfect | Non-profit |
| Q17: How do monopolies affect resources and welfare? | Fiscal | Ramsey | Monopoly | Commercial |
| Q18: What is algorithmic collusion in oligopolies? | Fiscal | Ramsey | Oligopoly | Non-profit |
| Q19: How does product diversity affect welfare? | Fiscal | Ramsey | Monopolistic | Non-profit |
| **Individual Decision Issues** | | | | |
| Q20: Does "996" work culture improve utility and efficiency? | * | OLG | Perfect | Non-profit |
| Q21: How does age affect consumption patterns? | * | OLG | Perfect | Commercial |
| Q22: How does asset allocation affect wealth? | * | Ramsey | Perfect | Commercial |
| Q23: How does work-life balance impact well-being? | * | Ramsey | Perfect | Non-profit |
| Q24: How does over-leveraged consumption impact the economy? | * | OLG | Perfect | Commercial |
| Q25: How do market structures shape consumer behavior? | * | Ramsey | All | Non-profit |

*Note*: * denotes any type. Abbreviations: Perfect = Perfect Competition, Monopolistic = Monopolistic Competition.
We provide a user manual for **EconGym** for these 25 economic problems, including problem descriptions, recommended economic roles and agent algorithms, and simple experiments. See: https://github.com/Miracle1207/EconGym.

Table 3: Core agent types in EconGym. Each type captures distinct modeling features and aligns with specific scenarios, enabling flexible composition of diverse economic simulations.

| Roles | Role Type | Model Features | Typical Scenarios |
|---|---|---|---|
| **Individual** | Ramsey model | Households, no aging | Wealth distribution, long-term dynamics |
| | OLG model | Age-specific agents, lifecycle dynamics | Retirement policy, demographic shifts |
| **Government** | Fiscal Authority | Sets tax and spending policy | Fiscal policy, redistribution |
| | Central Bank | Controls interest and reserve rates | Monetary policy, inflation control |
| | Pension Authority | Manages retirement age and pensions | Aging society, pension system reform |
| **Bank** | Non-profit Platform | Lending rate = deposit rate, no profit | Simplified setting |
| | Commercial Bank | Profit-maximizing under policy constraints | Policy impact on financial markets |
| **Firm** | Perfect Competition | Price-taking, market-clearing | Equilibrium analysis |
| | Monopoly | Single firm sets prices freely | Market power, regulation |
| | Oligopoly | Strategic pricing with few competitors | Cournot, collusion, AI pricing |
| | Monopolistic Comp. | Many firms, differentiated products | Branding, pricing strategy |

## 3.1 Economic Modeling

Economic research involves a wide range of complex and interdependent problems, each associated with distinct economic activities and decision processes. This inherent complexity poses challenges for building scalable economic simulation platforms, yet also presents opportunities for AI research. To ground our design, we compile a diverse set of 25 representative economic problems, listed in the first column of Table 2. These include canonical tasks such as optimal taxation (Q6), as explored by the AI Economist [47] and TaxAI [27], and labor supply incentives (Q20) in ABIDES-Economist [12].

Modeling each problem independently is prohibitively costly and limits generalization. Instead, EconGym adopts a unified structure centered on four foundational economic roles—**individuals, governments, firms, and banks**—which serve as universal building blocks across all scenarios. For example, individual decision-making underpins most economic activities, while government policy shapes behavior across all roles, from household consumption to firm pricing and bank interest setting. By modeling these shared roles, EconGym reduces scenario-specific design overhead. For each role, we implement multiple heterogeneous role types to accommodate diverse scenario requirements. Through composable combinations of role types and a unified interface for inter-role interactions, EconGym enables scalable construction of a wide range of economic tasks. A complete taxonomy of roles and their types is summarized in Table 3, with full model specifications in Appendix C.

**Individuals.** Individuals are microeconomic agents who make sequential decisions on consumption, labor supply, savings, and investment. These decentralized behaviors jointly determine macroeconomic outcomes such as income distribution and capital accumulation.

EconGym implements two representative types of individuals, each suited for different policy analyses: **(1)Ramsey agents** are infinitely-lived households facing idiosyncratic income shocks and incomplete markets. They are ideal for studying precautionary savings, asset accumulation, and household responses to taxation and monetary policy. **(2) Overlapping Generations (OLG) agents** capture lifecycle dynamics between working-age (Young) and retired (Old) individuals, enabling analysis of intergenerational equity, demographic shifts, and long-horizon fiscal interventions. Detailed mathematical formulations are in Appendix C.1.

Individuals interact with other agents through three core channels: **(1) Government:** They pay taxes and receive transfers under fiscal rules. OLG agents additionally contribute to or draw from pension systems (Appendix C.2). **(2) Banks:** Individuals deposit income or repay debt, with interest rates endogenously set by banking behavior (Appendix C.3). **(3) Firms:** They supply labor in exchange for wages and purchase goods for consumption. Prices and wages are determined by firm-side optimization under varying market structures (Appendix C.4).

**Government.** Government agents are institutional actors responsible for macro-level policy interventions. EconGym implements three representative types, each aligned with a distinct policy domain: **(1) Fiscal Authority** sets tax policy and spending, shaping production, consumption, and redistribution—suitable for studying optimal taxation and fiscal transfers. **(2) Central Bank** adjusts nominal interest rates and reserve requirements, transmitting monetary policy to households and firms. It supports scenarios focused on inflation control, monetary shocks, and stabilization. **(3)**

**Pension Authority** manages intergenerational transfers by setting retirement age, contribution rates, and pension payouts. It enables evaluation of long-run pension sustainability and demographic policy.

Government agents interact with individuals, firms, and banks through fiscal, monetary, and pension policies. These mechanisms collectively influence income distribution, firm behavior, and banking dynamics. Detailed formulations are provided in Appendix C.2.

**Banks.**  Banks serve as financial intermediaries that channel funds between individuals, firms, and governments. In EconGym, they support liquidity provision, capital allocation, and monetary transmission, especially through interest rates and bond markets.

We implement two distinct types of banking roles: **(1) Non-Profit Platforms** apply a uniform interest rate to deposits and loans, eliminating arbitrage and profit motives. They are widely used in economics for problem simplification. **(2) Commercial Banks** strategically set deposit and lending rates to maximize profits, subject to central bank constraints. These agents support macro-financial analysis involving credit supply, reserve constraints, and interest rate transmission. Banks interact with individuals via deposits and loans, with firms through business lending, and with governments through bonds and regulation. See Appendix C.3 for details.

**Firms.**  Firms are production agents that convert labor and capital into goods, influencing output, prices, wages, and income distribution. EconGym implements four firm types to support a variety of market structures: **(1) Perfectly Competitive Firms** are price takers with no strategic behavior, ideal for baseline analyses. **(2) Monopoly Firms** set prices and wages to maximize profits under aggregate demand constraints. **(3) Oligopoly Firms** engage in strategic competition, anticipating household responses and rival actions. **(4) Monopolistic Competitors** offer differentiated products with CES demand and endogenous entry, supporting studies of consumer preference and market variety.

These firm types enable scenario-specific modeling, from simplified environments for policy evaluation to complex settings with pricing power, strategic interactions, and market distortion. Firms interact with individuals as employers and producers, with banks through capital financing, and with governments via taxes, subsidies, and procurement. See Appendix C.4 for details.

## 3.2 Agent Modeling

Building on the economic roles described above, EconGym models each heterogeneous role type as a distinct agent in a Markov game. For each agent, we define a role-specific observation space, action space, and reward function. Given its private observation, each agent selects actions and receives rewards, while environment transitions are defined by economic mechanisms in Appendix C. This setup allows heterogeneous agents to make decisions and interact in diverse economic scenarios. Table 4 summarizes the observation, action, and reward design for all agent types in EconGym, with detailed Markov game formulations provided in Appendix C.1.1, C.2.1, C.3.1, and C.4.1.

Table 4: MDP Elements for Economic Agents in EconGym. For each agent, we summarize the observation $o_t$, action $a_t$, and reward $r_t$. Notation and economic meaning are annotated for clarity.

| Category | Variant | Observation $o_t$ | Action $a_t$ | Reward $r_t$ |
|---|---|---|---|---|
| **Individual** | **Ramsey model** | $o_t^i = (a_t^i, e_t^i)$ Private: assets, education, *OLG adds* $age_t^i$ Global: wealth distribution, education distribution, wage, price, lending rate, deposit rate | $a_t^i = (\alpha_t^i, \lambda_t^i, \theta_t^i)$ Asset allocation, labor, investment *OLG:* old agents $\lambda_t^i = 0$ | $r_t^i = U(c_t^i, h_t^i)$ (CRRA utility) *OLG includes pension if retired* |
| | **OLG model** | | | |
| **Government** | **Fiscal Authority** | $o_t^{\text{fiscal}} = \{\mathcal{A}_t, \mathcal{E}_{t-1}, W_{t-1}, P_{t-1}, r_{t-1}^l, r_{t-1}^d, B_{t-1}\}$ Wealth distribution, education distribution, wage rate, price level, lending rate, deposit rate, debt. | $a_t^{\text{fiscal}} = \{\boldsymbol{\tau}, G_t\}$ Tax rates, spending | GDP growth, equality, welfare |
| | **Central Bank** | $o_t^{\text{cb}} = \{\mathcal{A}_t, \mathcal{E}_{t-1}, W_{t-1}, P_{t-1}, r_{t-1}^l, r_{t-1}^d, \pi_{t-1}, g_{t-1}\}$ Wealth distribution, education distribution, wage rate, price level, lending rate, deposit rate, inflation rate, growth rate. | $a_t^{\text{cb}} = \{\phi_t, \iota_t\}$ Reserve ratio, benchmark rate | Inflation/GDP stabilization |
| | **Pension Authority** | $o_t^{\text{pension}} = \{F_{t-1}, N_t, N_t^{old}, age_{t-1}^r, \tau_{t-1}^p, B_{t-1}, Y_{t-1}\}$ Pension fund, current population, old individuals' number, last retirement age, last contribution rate, debt, GDP | $a_t^{\text{pension}} = \{age^r, \tau_p, k\}$ Retirement age, contribute, growth | Pension fund sustainability |
| **Bank** | **Non-Profit Platform** | / | No rate control | No profit |
| | **Commercial Bank** | $o_t^{\text{bank}} = \{\iota_t, \phi_t, r_{t-1}^l, r_{t-1}^d, \text{loan}, F_{t-1}\}$ Benchmark rate, reserve ratio, last lending rate, last deposit rate, loans, pension fund. | $a_t^{\text{bank}} = \{r_t^d, r_t^l\}$ Deposit, lending decisions | $r = r_t^l(K_{t+1} + B_{t+1}) - r_t^d A_{t+1}$ Interest margin |
| **Firm** | **Perfect Competition** | / | / | Zero (long-run) |
| | **Monopoly** | $o_t^{\text{mono}} = (K_t, Z_t, r_{t-1}^l)$ Production capital, productivity, lending rate | $a_t^{\text{mono}} = (p_t, W_t)$ Price and wage decisions | $r_t^{\text{mono}} = p_t Y_t - W_t L_t - R_t K_t$ Profits = Revenue − costs |
| | **Oligopoly** | $o_t^{\text{olig}} = (K_t^j, Z_t^j, r_{t-1}^l)$ Production capital, productivity, lending rate | $a_t^{\text{olig}} = (p_t^j, W_t^j)$ Price and wage decisions for firm $j$ | $r_t^{\text{olig}} = p_t^j y_t^j - W_t^j L_t^j - R_t K_t^j$ Profits = Revenue − costs for firm $j$ |
| | **Monopolistic Competition** | $o_t^{\text{mono-comp}} = (K_t^j, Z_t^j, r_{t-1}^l)$ Production capital, productivity, lending rate. Here, $j$ denotes the firm index. | $a_t^{\text{mono-comp}} = (p_t^j, W_t^j)$ Price and wage decisions for firm $j$ | $r_t^{\text{mono-comp}} = p_t^j y_t^j - W_t^j L_t^j - R_t K_t^j$ Profits = Revenue − costs for firm $j$ |

# 4 Experiments

To assess EconGym's abilities in economic analysis and AI policy learning, we design 3 experiments:

1. **Single-Scenario Modeling: Which AI Agents Best Govern Pension Policy?** We simulate long-term aging dynamics under varied retirement ages, benchmarking four pension agents: RL, LLM, rule-based, and real data policy.

2. **Cross-domain Tasks: Are Multi-Government Settings Better? Can AI Help?** We compare isolated (monetary, fiscal, pension) versus coordinated (e.g., fiscal + monetary + pension) policy setups, evaluating AI benchmarks under multi-government coordination.

3. **Realism vs. Efficiency: Does Scaling AI to Larger Populations Pay Off?** We assess trade-offs between simulation fidelity and efficiency across population sizes and model complexities.

We further validate EconGym against established economic benchmark models and peer simulation platforms, with detailed results provided in Appendix D.

## 4.1 Overview of Agent Algorithms: Matching Algorithms to Economic Problems.

Different economic problems often require different decision-making paradigms. To support this, EconGym accommodates six types of agent algorithms:

- **Reinforcement Learning (RL)** agents learn through trial-and-error to optimize long-term cumulative rewards. RL is well-suited for solving optimal decision-making problems in dynamic environments.

- **Large Language Models (LLMs)** generate decisions based on internal knowledge and language understanding. While not always optimal, LLMs often exhibit human-like behavior patterns, making them valuable for simulating realistic decision-making.

- **Behavior Cloning (BC)** agents imitate real-world behavior by training on empirical data. In our experiments, individual households follow BC policies learned from the 2022 Survey of Consumer Finances [2] data, enabling realistic micro-level behavior.

- **Economic Method** refers to classical rule-based policies from economics literature—for example, using the Taylor rule [42] for central banks or Saez Tax [38] for fiscal agents. These allow direct comparisons between economic theory and AI-based approaches.

- **Expert Rules** encode domain knowledge or user-defined heuristics—for instance, the IMF's fiscal adjustment rule [13] or informal advice like "save more when young for retirement". These rules offer interpretable, human-crafted policies.

- **Real-Data** policies replay actual policy trajectories based on historical data, such as U.S. federal tax rates or retirement age schedules. This enables direct benchmarking against real-world policy outcomes.

Each algorithm has its own strengths. EconGym supports benchmarking them under the same economic role, or combining different algorithms across roles in a shared scenario. In the following experiments, we showcase how these algorithms generate diverse policy outcomes across tasks.

## 4.2 Single-Scenario Modeling: Which AI Agents Best Govern Pension Policy?

**Unveiling Aging Dynamics: How Retirement Age Shapes Economic Sustainability?** We explore the *aging-pension* scenario to demonstrate EconGym's capability in simulating long-term population shifts, macroeconomic indicators, and the effects of delayed retirement. In this task, the government selects the pension authority; individuals (N=1000) use the OLG model and a behavior cloning policy trained on 2022 Survey of Consumer Finances data, incorporating U.S. birth rates [31] and CDC's 2022 stepwise mortality rates [1] (see Appendix Table 8). Other economic roles are freely configurable. With 1,000 individuals, EconGym simulates a declining population and labor force over time in Fig.3(a). The median age shifts from 45–65 to 55–85 over 60 years, capturing aging dynamics in Fig.3(b).

We assess five retirement ages (60, 63, 65, 67, 70) and track key outcomes: GDP, consumption, social welfare, utility, pension fund balance, and dependency ratio. As shown in Fig.3(c–h), GDP and consumption rise initially—signaling short-term sustainability—but decline after 20 years due to

---

[2]https://www.federalreserve.gov/econres/scfindex.htm

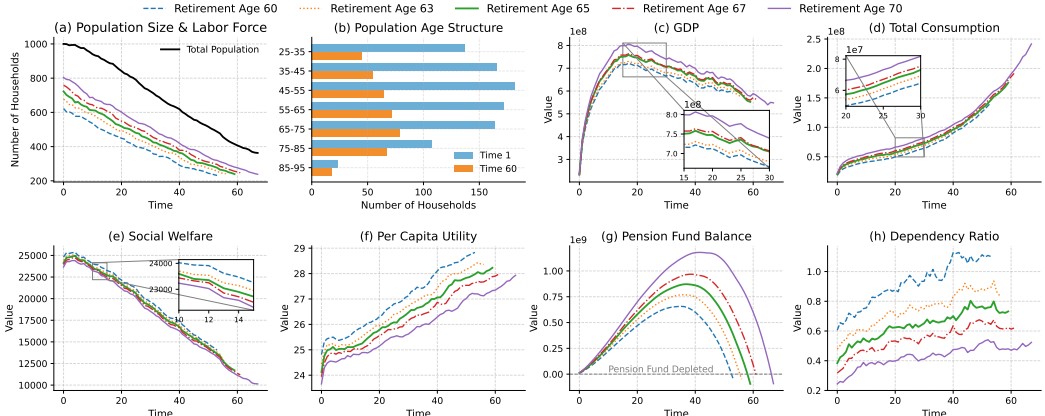

Figure 3: Evolution of economic indicators under varying retirement ages (60, 63, 65, 67, 70) in the aging-pension scenario: (a) population and labor force decline, (b) age structure shifts toward older demographics, (c-h) GDP, consumption, social welfare, per capita utility, pension fund balance, and dependency ratio trends, illustrating the impact of delayed retirement on economic indicators and individual welfare.

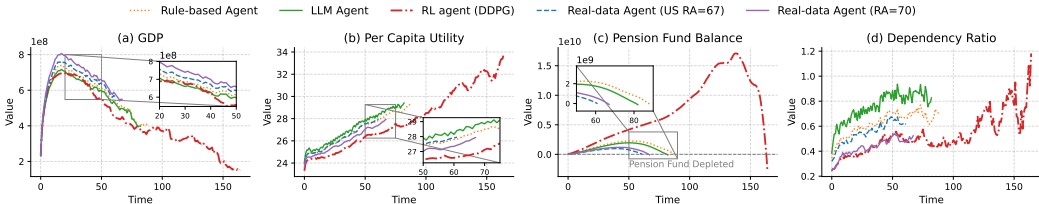

Figure 4: Performance of agent algorithms (RL, LLM, rule-based, real-data) in pension policy optimization, showcasing RL's superior economic sustainability (165 years), LLM's highest individual welfare, and real-data agent's early GDP peak but earliest pension fund depletion.

aging. Higher retirement ages extend labor participation, boosting GDP and consumption (Fig.3(a)). Pension sustainability also improves: retiring at 70 delays fund depletion by 15 years compared to age 60 (Fig.3(g)) and reduces the dependency ratio (Fig.3(h)). However, later retirement lowers social welfare and per capita utility (Fig.3(e, f)), reflecting trade-offs aligned with economic intuition.

**Battle of AI Strategies: Which Agent Masters Pension Policy Optimization?** In aging task, we benchmark four agent types—LLM, RL, rule-based, and real-data—as pension authorities aiming to optimize fund longevity and long-term population welfare. The rule-based agent uses the IMF adjustment rule [13]. Figure 4 shows that RL agents excel in long-term optimization, sustaining pension funds for 165 years—nearly 100 years longer than the real-data agent (RA=67). Though their GDP and utility may lag at specific points, RL agents achieve the highest total GDP and utility over time, with the lowest dependency ratio. LLM agents emphasize human-centric strategies, delivering high per capita utility (Fig.4(b)) but shorter fund longevity. Real-data agents start with strong GDP but deplete funds fastest (Fig.4(a)). Rule-based agents strike a middle ground—more sustainable than real-data agents but trailing RL in longevity and LLM in utility. Overall, each agent type shows distinct strengths, with RL agents emerging as the most effective for complex policy optimization. Representative outputs of the LLM policy agent, illustrating its reasoning and adaptation process, are provided in Appendix E.

### 4.3 Cross-domain Tasks: Are Multi-Government Settings Better? Can AI Help?

**Breaking Policy Silos: Multi-Government Interactions Uncover Synergy and Conflict.** In the multi-government coordination task, we jointly simulate multiple government agents solving policies within a single scenario. We evaluate six combinations of fiscal, central bank, and pension authorities, using 1,000 individuals with BC policies. The fiscal agent adopts the Saez tax [38], the central bank follows the Taylor rule [42], and the pension agent implements the IMF retirement adjustment

rule [13]. Results in Fig. 5 highlight three key insights: **(1) Coordination yields synergy.** As shown in Fig.5(a), *Fiscal Only* (green) and *Central Bank Only* (orange) achieve similar GDP peaks but decline rapidly. Their combination—*Fiscal + Central Bank* (red)—extends economic longevity, increases fiscal revenue (Fig.5(d)), and reduces inequality (Fig. 5(c)). **(2) Combining independently designed policies may cause conflict.** *Fiscal + Pension* (blue) improves short-term welfare (Fig.5(b)) but underperforms *Fiscal Only* in GDP and lifespan (Fig.5(a)). The full combination—*Fiscal + Central Bank + Pension* (purple)—yields the weakest overall performance, indicating that uncoordinated policies can generate harmful dynamics. **(3) AI agents enable adaptive synergy.** Replacing the pension rule with an RL agent—*Fiscal + Central Bank + Pension (RL)* (black)—achieves the best overall results: higher GDP, stronger revenue, lower inequality, and stable welfare. This demonstrates the potential of adaptive AI in orchestrating complex, multi-agent policy environments.

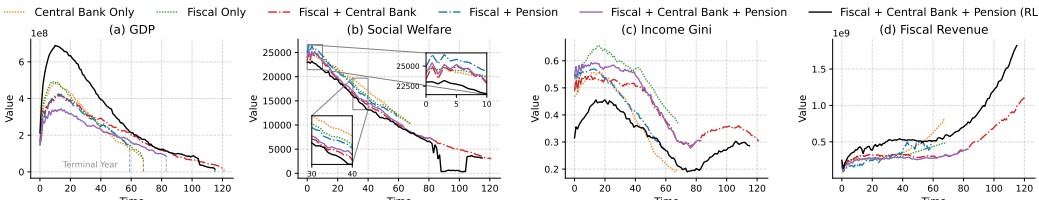

Figure 5: Simulation outcomes under 6 heterogeneous combinations of fiscal, central bank, and pension authority. We observe both synergistic effects (e.g., *Fiscal + Central Bank*) and conflicting dynamics (e.g., *Fiscal + Central Bank + Pension*) across GDP, welfare, inequality, and revenue. Replacing the economic method of pension agent with RL (*Fiscal + Central Bank + Pension (RL)*) significantly improves outcomes, demonstrating the adaptability of AI agents for policy coordination.

Table 5: Benchmark results across multi-government coordination tasks with varying algorithmic combinations in EconGym. This is a condensed summary; full version appear in Appendix Table 6.

| ID | Fiscal Gov. | Pension Gov. | Central Bank | Year | Consumption | Avg. Labor | GDP | Welfare | Gini |
|----|-------------|--------------|--------------|------|-------------|------------|--------|---------|------|
| 2  | –           | –            | Taylor rule  | 73   | 9.48e+07    | 1053.80    | 2.32e+10 | 1.26e+06 | 0.37 |
| 4  | –           | IMF          | –            | 76   | 7.44e+07    | 1051.10    | 2.27e+10 | 1.24e+06 | 0.37 |
| 6  | Saez Tax    | –            | –            | 68   | 1.18e+08    | 1047.05    | 2.34e+10 | 1.19e+06 | 0.48 |
| 7  | Real data   | –            | Real data    | 73   | 9.46e+07    | 1054.00    | 2.33e+10 | 1.26e+06 | 0.36 |
| 9  | Real data   | Real data    | –            | 69   | 8.88e+07    | 1117.20    | 2.45e+10 | 1.21e+06 | 0.37 |
| 21 | LLM         | LLM          | LLM          | 7    | 3.78e+06    | 744.45     | 1.98e+09 | 8.42e+04 | 0.68 |
| 22 | PPO         | PPO          | PPO          | 5    | 2.21e+04    | 1019.57    | 1.50e+09 | -8.91e+04 | 0.37 |
| 27 | Saez Tax    | DDPG         | DDPG         | 79   | 4.74e+07    | 1340.95    | 2.63e+10 | 1.19e+06 | 0.33 |
| 28 | Saez Tax    | DDPG         | LLM          | 78   | 9.35e+07    | 1354.60    | 3.06e+10 | 1.17e+06 | 0.32 |

**AI-Economic Strategies Win: AI Benchmarking under Multi-Government Coordination**  We benchmark 33 configurations spanning LLMs, RL, economic, and real-data policies across single, double, and multi-government tasks. Table 5 reveals three key findings: **(1)** Multi-government tasks expand the optimization space. Adding government agents improves macroeconomic outcomes via expanded policy space and agent collaboration such as `ID-7` and `ID-9`. **(2)** Pure AI policies often underperform in complex economic settings. Without structural priors, LLM-only (`ID-21`) or RL-only (`ID-22`) agents struggle with the high-dimensional, multi-agent environment. **(3)** Hybrid approaches (AI + economics) consistently deliver superior results. Combining economic rules with adaptive AI yields the highest GDP, welfare, and the lowest inequality such as `ID-27, ID-28`. Together, these results position EconGym as a unified testbed for AI, economic and hybrid methods in complex economic problems. See Appendix A.1 for full results Table 6 and analysis.

## 4.4    Realism vs. Efficiency: Does Scaling AI to Large Populations Pay Off?

**Simulation Realism Improves with Larger Populations.**  We examine how scaling up individual agents in EconGym affects simulation realism under real-world policy settings: U.S. 2022 federal progressive taxes (fiscal), real interest rates (monetary), and a retirement age of 67 (pension), without external interventions. We assess realism by comparing simulated population distributions to the 2022 Survey of Consumer Finances. Figure 6 presents consumption patterns, labor supply by age, and Lorenz curves for income and wealth across population sizes ($N = 10$ to $100,000$), with real

data in black. As $N$ increases, simulated outcomes better match empirical distributions. Notably, Figure 6(a,b) reproduces the classic "hump-shaped" consumption and "inverted U-shaped" labor supply curves, consistent with prior studies [14, 24]. To quantify realism, we compute the Wasserstein Distance (WD) between simulated and real distributions, where lower WD indicates higher fidelity. As shown in Figure 6, WD consistently declines across consumption, labor, age, and wealth as $N$ grows, validating that larger populations enhance realism in EconGym.

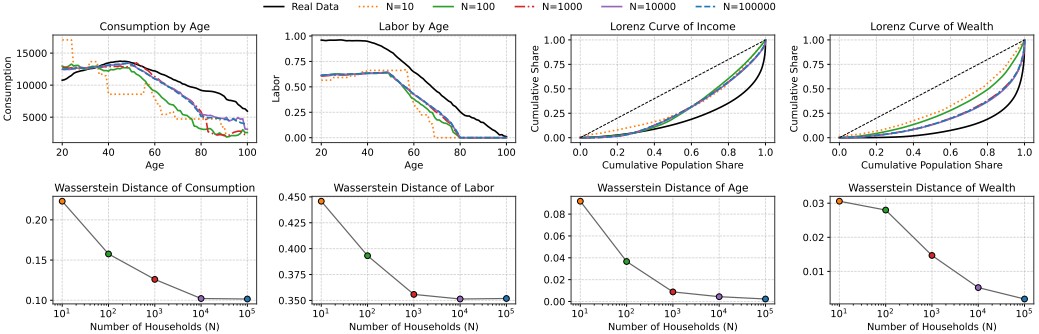

Figure 6: Realism evaluation across scales (N=10 to 100,000): Top row compares consumption by age, labor by age, and Lorenz curves for income and wealth against real data (black). Bottom row shows Wasserstein Distance (WD) for consumption, labor, age, and wealth distributions, decreasing with larger N, confirming higher fidelity to real distributions.

**Sampling Efficiency of EconGym.** As the number of individuals $N$ increases from 10 to 1 million, the step time in **EconGym** grows accordingly. We benchmark the performance on an Apple M1 Pro (16GB) and report the results for 25 tasks in Fig. 7 and Table 7 (Appendix A.2). In Fig. 7, we plot the average step time (red line) across all 25 tasks. The colored dashed lines represent 25 tasks, each exhibiting distinct step times due to differences in their underlying economic dynamics and modeling complexity. Nevertheless, all tasks follow a consistent overall trend that aligns with the mean value. At $N = 100$k, the average environment step time is approximately **32.125 ms**, indicating efficient large-scale simulation performance. **EconGym** thus recommends simulating between 100 and 100k dynamically interacting agents to achieve a favorable balance between realism and sampling efficiency.

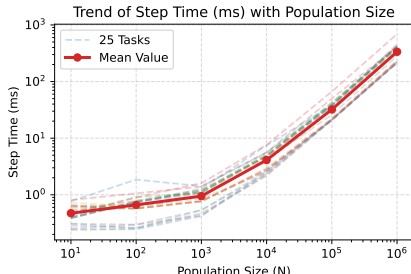

Figure 7: Step time across scales ($N = 10 \sim 10^6$): mean step time (solid line) and 25 tasks (dashed lines).

## 5 Discussion and Conclusion

**Key findings** from our experiments: (1) EconGym enables fine-grained modeling in both single-domain (e.g., aging) and cross-domain tasks. Combining independently designed policies can yield synergy or conflict, which adaptive AI agents learn to handle. (2) EconGym benchmarks hybrid configurations of AI, economic methods, and real data, showing that combining AI with structured economics consistently outperforms pure learning-based approaches in complex settings. (3) EconGym maintains both accuracy and efficiency as the population scales.

These findings stem from EconGym's **core design**: it integrates a rich set of economic models, formally defining agent structures and environment transitions. This enables the composition of diverse environment with heterogeneous agents, where dynamic interactions yield data that support both AI research and economic analysis. Yet, **this is only the beginning.** Future work will expand agent populations, diversify tasks, and incorporate richer empirical data—further contributing to the integration of AI and economics.

## Acknowledgments

We sincerely thank Prof. Bo Li from Peking University for his insightful discussions and constructive feedback throughout the development of this project. As an accomplished economist, Prof. Li provided crucial guidance on the theoretical underpinnings of the economic models in EconGym, greatly enhancing the platform's rigor and realism.

This work was supported in part by the National Natural Science Foundation of China under the Original Exploration Program (Grant No. 72450002).

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

# A    Additional Experimental Results

Table 6: Simulation results under varying numbers of government agents in EconGym. The table is sorted by the number of active government agents and algorithm type.

| ID | Fiscal Gov. | Pension Gov. | Central Bank | Year | Consumption | Avg. Labor | GDP | Welfare | Gini |
|---|---|---|---|---|---|---|---|---|---|
| 1 | – | – | Real data | 72 | 8.89e+07 | 1052.90 | 2.23e+10 | 1.21e+06 | 0.37 |
| 2 | – | – | Taylor rule | 73 | 9.48e+07 | 1053.80 | 2.32e+10 | 1.26e+06 | 0.37 |
| 3 | – | Real data | – | 69 | 8.62e+07 | 1116.03 | 2.34e+10 | 1.17e+06 | 0.37 |
| 4 | – | IMF | – | 76 | 7.44e+07 | 1051.10 | 2.27e+10 | 1.24e+06 | 0.37 |
| 5 | Real data | – | – | 72 | 8.89e+07 | 1052.90 | 2.23e+10 | 1.21e+06 | 0.37 |
| 6 | Saez Tax | – | – | 68 | 1.18e+08 | 1047.05 | 2.34e+10 | 1.19e+06 | 0.48 |
| 7 | Real data | – | Real data | 73 | 9.46e+07 | 1054.00 | **2.33e+10** | 1.26e+06 | 0.36 |
| 8 | Real data | – | Taylor rule | 71 | 9.10e+07 | 1048.34 | 2.20e+10 | 1.23e+06 | 0.38 |
| 9 | Real data | Real data | – | 69 | 8.88e+07 | 1117.20 | **2.45e+10** | 1.21e+06 | 0.37 |
| 10 | Real data | IMF | – | 66 | 8.40e+07 | 1057.24 | 2.27e+10 | 1.19e+06 | 0.38 |
| 11 | Saez Tax | – | Real data | 77 | 2.55e+07 | 1040.04 | 1.83e+10 | 1.22e+06 | 0.38 |
| 12 | Saez Tax | – | Taylor rule | 54 | 1.33e+08 | 1067.64 | 1.94e+10 | 1.05e+06 | 0.41 |
| 13 | Saez Tax | Real data | – | 75 | 5.50e+07 | 1108.28 | 2.26e+10 | 1.22e+06 | 0.37 |
| 14 | Saez Tax | IMF | – | 75 | 4.22e+07 | 1044.29 | 2.25e+10 | 1.23e+06 | 0.38 |
| 15 | Real data | Real data | Real data | 78 | 6.58e+07 | 1157.46 | 2.41e+10 | 1.22e+06 | 0.39 |
| 16 | Real data | DDPG | Taylor rule | 55 | 5.63e+07 | 698.80 | 1.39e+10 | 1.13e+06 | 0.49 |
| 17 | DDPG | Real data | Taylor rule | 50 | 2.35e+08 | 1138.91 | 2.02e+10 | 9.85e+05 | 0.40 |
| 18 | DDPG | DDPG | DDPG | 2 | 3.09e+03 | 1381.91 | 4.15e+08 | -1.04e+05 | 0.00 |
| 19 | DDPG | DDPG | Taylor rule | 50 | 2.85e+08 | 1389.79 | 2.47e+10 | 9.47e+05 | 0.36 |
| 20 | LLM | DDPG | Taylor rule | 9 | 5.12e+08 | 1316.10 | 4.67e+09 | -5.26e+04 | 0.37 |
| 21 | LLM | LLM | LLM | 7 | 3.78e+06 | 744.45 | 1.98e+09 | 8.42e+04 | 0.68 |
| 22 | PPO | PPO | PPO | 5 | 2.21e+04 | 1019.57 | 1.50e+09 | -8.91e+04 | 0.37 |
| 23 | PPO | PPO | PPO | 54 | 5.70e+07 | 625.97 | 1.18e+10 | 9.96e+05 | 0.51 |
| 24 | Saez Tax | Real data | DDPG | 55 | 1.41e+08 | 1129.62 | 2.08e+10 | 1.05e+06 | 0.40 |
| 25 | Saez Tax | Real data | Taylor rule | 55 | 1.41e+08 | 1129.62 | 2.08e+10 | 1.05e+06 | 0.40 |
| 26 | Saez Tax | DDPG | Real data | 49 | 8.68e+07 | 636.10 | 1.12e+10 | 1.05e+06 | 0.52 |
| 27 | Saez Tax | DDPG | DDPG | 79 | 4.74e+07 | 1340.95 | **2.63e+10** | 1.19e+06 | 0.33 |
| 28 | Saez Tax | DDPG | LLM | 78 | 9.35e+07 | 1354.60 | **3.06e+10** | 1.17e+06 | 0.32 |
| 29 | Saez Tax | DDPG | PPO | 49 | 8.68e+07 | 636.10 | 1.12e+10 | 1.05e+06 | 0.52 |
| 30 | Saez Tax | DDPG | Taylor rule | 82 | 5.94e+06 | 605.74 | 1.10e+10 | 9.82e+05 | 0.51 |
| 31 | Saez Tax | LLM | Taylor rule | 60 | 5.36e+07 | 924.01 | 1.92e+10 | 1.09e+06 | 0.43 |
| 32 | Saez Tax | PPO | Taylor rule | 54 | 1.31e+08 | 998.11 | 1.85e+10 | 1.06e+06 | 0.43 |
| 33 | Saez Tax | IMF | Taylor rule | 84 | 1.52e+07 | 886.44 | 1.84e+10 | 1.26e+06 | 0.47 |

## A.1    Hybrid AI-Economic Strategies Win: AI Benchmarking under Multi-Government Coordination

We evaluate algorithm performance across increasingly complex *Multi-Government coordination* tasks—*single*, *double*, and *multi-government*—featuring different combinations of fiscal, monetary, and pension authorities. Our benchmark spans 33 configurations, including reinforcement learning (RL), large language models (LLMs, e.g., `DeepSeek-V3-0324`), real-world policy data (e.g., U.S. retirement age = 67), and classic economic rules (e.g., Saez tax, Taylor rule). Results are summarized in Table 6. We highlight three key findings from these experiments:

**1. Multi-government structures unlock richer coordination opportunities.** Adding government agents expands the policy space and enables more inter-agent collaboration. In several double-government scenarios, such as `ID-7` and `ID-9` (GDP: **2.45e+10**, Welfare: **1.21e+06**), we observe notable improvements over single-government baselines like `ID-1` ~ `ID-6` (GDP: **2.23e+10**, Welfare: **1.21e+06**). This demonstrates that multi-government configurations can achieve better macroeconomic outcomes by enabling richer coordination across institutional agents.

**2. Pure AI policies often underperform in complex economic settings.** AI-only agents frequently struggle to find effective strategies. For instance, `ID-21` (LLM-only) results in GDP: **1.98e+09** and Welfare: **8.42e+04**; while `ID-22` (PPO-only) terminates with GDP: **1.50e+09**, Welfare: **-8.91e+04**. These failures stem from the high-dimensional decision space created by heterogeneous agents with asymmetric observations. Without domain priors or structural guidance, AI agents struggle to discover effective strategies. This underscores the need for AI algorithms specifically designed for economic environments with multi-agent complexity.

**3. Hybrid approaches (AI + economics) consistently deliver superior results.** Top-performing configurations integrate structured economic rules with adaptive AI agents. For example, `ID-28`

(Saez tax + DDPG + LLM) achieves the highest GDP: **3.06e+10**, strong Welfare: **1.17e+06**, and the lowest Gini: **0.32** among all tested setups. Other hybrid setups, such as `ID-27` (GDP: **2.63e+10**, Gini: **0.33**), also demonstrate robust performance. These combinations constrain the policy space using domain knowledge, while allowing AI components to optimize within structured boundaries. Together with Finding 2, this highlights the importance of embedding economic priors into AI policy design to navigate complex, high-dimensional decision environments effectively.

**Takeaway: EconGym offers a rigorous benchmark for AI-driven economic governance.** By supporting heterogeneous agents, modular policy roles, and real-world-inspired settings, **EconGym** provides a high-fidelity testbed for evaluating AI algorithms under structured, multi-agent economic dynamics. It reveals clear limitations of current methods and motivates the development of structure-aware, compositional AI frameworks.

## A.2 Efficiency

As shown in Table 7, we evaluated the environment sampling step times for the 25 economic tasks presented in this paper across varying population sizes (N=10 to 1,000,000). The names of these tasks correspond to the abbreviations of the economic problems listed in Table 2, and they also serve as the names of the parameter files for each task in our GitHub repository (for more details, please refer to our GitHub repo: `https://github.com/Miracle1207/EconGym` ). Additionally, we report the mean step times across all tasks, which are presented in the final row of the table.

Table 7: Step time (ms) at varying population sizes (N).

| Tasks (YAML Name) | N=10 | N=100 | N=1,000 | N=10,000 | N=100,000 | N=1,000,000 |
|---|---|---|---|---|---|---|
| delayed_retirement | 0.415 | 0.743 | 1.284 | 7.434 | 38.021 | 428.524 |
| personal_pension | 0.390 | 0.782 | 1.207 | 4.841 | 35.478 | 398.457 |
| population_aging | 0.389 | 0.727 | 1.193 | 4.838 | 41.252 | 412.033 |
| pension_gap | 0.438 | 0.899 | 1.606 | 7.641 | 66.533 | 692.047 |
| pension_across_countries | 0.389 | 0.746 | 1.121 | 4.809 | 40.473 | 400.084 |
| consumption_tax | 0.307 | 0.297 | 0.538 | 2.369 | 21.308 | 223.003 |
| estate_tax | 0.406 | 0.743 | 1.073 | 4.793 | 37.928 | 457.300 |
| universal_basic_income | 0.384 | 0.752 | 1.207 | 4.846 | 38.486 | 441.864 |
| optimal_tax | 0.238 | 0.256 | 0.473 | 2.182 | 21.137 | 241.364 |
| wealth_tax | 0.309 | 0.256 | 0.534 | 2.619 | 21.253 | 215.794 |
| negative_interest | 0.245 | 0.246 | 0.422 | 2.362 | 21.090 | 220.007 |
| inflation_control | 0.553 | 0.570 | 0.770 | 2.633 | 21.360 | 219.598 |
| QE | 0.658 | 0.566 | 0.780 | 2.833 | 20.592 | 216.561 |
| optimal_monetary | 0.626 | 0.573 | 0.765 | 2.776 | 21.396 | 216.744 |
| dbl_government | 0.284 | 0.262 | 0.458 | 2.191 | 20.885 | 212.618 |
| technology | 0.513 | 0.790 | 1.119 | 4.612 | 37.385 | 378.126 |
| monopoly | 0.593 | 0.670 | 0.848 | 2.580 | 21.840 | 218.916 |
| oligopoly | 0.628 | 0.581 | 0.985 | 4.069 | 25.975 | 282.529 |
| monopolistic_competition | 0.729 | 0.621 | 1.096 | 5.044 | 38.837 | 405.933 |
| work_hard | 0.391 | 0.754 | 1.136 | 5.433 | 36.568 | 413.765 |
| age_consumption | 0.773 | 1.857 | 1.418 | 5.658 | 38.316 | 411.893 |
| asset_allocation | 0.643 | 0.648 | 0.753 | 2.992 | 22.213 | 229.047 |
| work_life_well_being | 0.469 | 0.907 | 1.068 | 5.081 | 42.909 | 404.245 |
| over_leveraged_consumption | 0.809 | 1.061 | 1.394 | 5.595 | 50.099 | 414.194 |
| market_type | 0.255 | 0.297 | 0.426 | 2.595 | 21.794 | 230.198 |
| **Mean Value** | 0.473 | 0.664 | 0.947 | 4.113 | 32.125 | 335.394 |

## B Limitations and Future Directions

While EconGym provides a unified and scalable AI testbed for simulating diverse economic decision-making tasks, several limitations remain before real-world deployment becomes feasible.

**First**, the current version supports a core set of economic roles and benchmark tasks. Expanding to more complex institutions—such as local governments, insurance providers, international trade, and cross-country interactions—will be essential for capturing richer economic dynamics. **Second**, agent-

scale remains limited. Bridging toward population-level modeling, with millions of heterogeneous individuals, is a long-term goal necessary for high-fidelity macroeconomic simulations. **Third**, policy calibration remains incomplete. While EconGym supports calibrated individual behaviors using empirical distributions, government interventions are not yet tuned to real-world responses due to the scarcity of high-quality public datasets capturing both policy actions and downstream human responses. Partnering with institutions to unlock such datasets will be critical for further realism and deployment.

Together, these limitations underscore that EconGym is a foundational testbed—not an end solution. But by enabling rigorous AI benchmarking in economic contexts, it opens the door to future research that pushes closer to real-world impact.

# C  Full Mathematical Models of EconGym

## C.1  Individuals

In economic theory, an **individual** refers to a microeconomic agent who makes decisions regarding consumption, labor supply, savings, and investment [26, 44]. These decisions are typically guided by utility maximization under budgetary and institutional constraints. As individual behavior aggregates to shape macroeconomic outcomes, accurate individual modeling is central to economic analysis [5].

Our simulation environment models individuals as agents who choose labor supply, consumption, and allocate remaining wealth across savings and risky investments. To support a wide range of economic scenarios, we implement two canonical formulations: the **Ramsey model** and the **Overlapping Generations (OLG) model**. While both share a common utility structure, they differ in time horizons, demographic assumptions, and policy interactions. The Ramsey model captures infinite-horizon behavior with income uncertainty, whereas the OLG model focuses on lifecycle dynamics, intergenerational transfers, and pension systems.

All individuals maximize lifetime utility over a finite horizon $T$:

$$U = \sum_{t=0}^{T} \beta^t \left[ \frac{c_t^{1-\sigma}}{1-\sigma} - \frac{h_t^{1+\gamma}}{1+\gamma} \right],$$

where $c_t^i$ and $h_t^i$ denote consumption and labor supply of individual $i$ at time $t$. The discount factor $\beta \in (0,1)$ governs intertemporal preferences. Parameters $\sigma$ and $\gamma$ represent the coefficient of relative risk aversion and the inverse Frisch elasticity of labor supply, respectively.

**Ramsey model**  The Ramsey model features infinitely-lived households [2] facing idiosyncratic income shocks and incomplete markets. To capture portfolio choices, asset holdings $a_t^i$ are decomposed into low-risk savings $s_t^i$ and risky investments $v_t^i$. The household budget constraint is:

$$p_t c_t^i + s_{t+1}^i + v_{t+1}^i = (1 + r_{t-1})s_t^i + (1 + \rho_{t-1})v_t^i + W_t e_t^i h_t^i - T_t(r_{t-1}s_t^i, \rho_{t-1}v_t^i, W_t e_t^i h_t^i), \quad (1)$$

where $r_{t-1}$ and $\rho_{t-1}$ denote the risk-free and risky returns. $W_t$ is the wage rate, $e_t^i$ is the education level, and $T_t(\cdot)$ is the tax function.

Households control two financial decision variables: the *allocation ratio* $\alpha_t^i \in [0,1]$, indicating the share of resources allocated to financial planning (i.e., not consumed), and the *investment ratio* $\theta_t^i \in [0,1]$, which determines the portion invested in risky assets. The resulting next-period asset $a_{t+1}^i$ composed of savings and investments:

$$a_{t+1}^i = s_{t+1}^i + v_{t+1}^i = \alpha_t^i \cdot m_t^i,$$

where $m_t^i$ denotes total disposable resources in the current period after tax, defined as:

$$m_t^i = (1 + r_{t-1})s_t^i + (1 + \rho_{t-1})v_t^i + W_t e_t^i h_t^i - T_t(\cdot).$$

The allocation is split into savings and investment according to:

$$s_{t+1}^i = (1 - \theta_t^i) \cdot a_{t+1}^i, \quad v_{t+1}^i = \theta_t^i \cdot a_{t+1}^i,$$

with consumption determined residually by $\alpha_t^i = 1 - \frac{p_t c_t^i}{m_t^i}$. Households face income uncertainty and make saving, investment, labor, and consumption choices accordingly, resulting in precautionary saving behavior [10].

**Overlapping Generations (OLG) Model**   In the OLG model, individuals are classified as *Young* or *Old* based on a policy-defined retirement age. Individuals remain economically active (Young) until their age exceeds this statutory threshold, after which they transition into retirement (Old). This structure enables age-specific decision modeling and intergenerational policy analysis.

**Young individuals** participate in the labor market and contribute to pensions. Their budget constraint is:

$$p_t c_t^i + s_{t+1}^i + v_{t+1}^i = (1 + r_t)s_t^i + (1 + \rho_t)v_t^i + W_t e_t^i h_t^i - T_t(r_t s_t^i, \rho_t v_t^i, W_t e_t^i h_t^i) - f_t^y,$$

where $f_t^y$ denotes mandatory pension contributions. Financial decisions mirror those in the Ramsey model. The only difference lies in the budget composition: unlike the Ramsey model setting, total disposable resources $m_t^i$ are reduced by pension contributions $f_t^y$:

$$\text{where } m_t^i = (1 + r_t)s_t^i + (1 + \rho_t)v_t^i + W_t e_t^i h_t^i - T_t(\cdot) - f_t^y, \quad \alpha_t^i = 1 - \frac{p_t c_t^i}{m_t^i}.$$

**Old individuals** no longer work and rely on accumulated wealth and pension benefits. Their budget constraint is:

$$p_t c_t^i + s_{t+1}^i + v_{t+1}^i = (1 + r_t)s_t^i + (1 + \rho_t)v_t^i - T_t(r_t s_t^i, \rho_t v_t^i) + f_t^o,$$

where $f_t^o$ is the pension received.

Each period, age evolves as $\text{age}_t^i = \text{age}_{t-1}^i + 1$, with retirement at $\text{age}^r$ and death at $\text{age}_{\max}$. Population updates follow:

$$N_{t+1} = (1 + n_t^b - n_t^d)N_t, \quad x_{t+1} = \frac{(x_t + n_t^b - n_t^d)N_t}{N_{t+1}},$$

where $n_t^b$ and $n_t^d$ denote birth and death rates (e.g. follows Table 8), and $x_t$ is the share of young individuals.

At $t = 0$, attributes like age and education are initialized using real-world data [3]. Inheritance from the deceased defines the initial wealth of newborns:

$$a_{t+1}^i(\text{age} = 1) = \frac{1}{n_t^b N_t} \sum_{j \in \mathcal{N}_t^d} (s_t^j + v_t^j),$$

where $\mathcal{N}_t^d$ is the set of deceased individuals. This intergenerational transfer mechanism ensures capital continuity across generations.

By modeling age progression, financial heterogeneity, and demographic turnover, the OLG model provides a flexible foundation for evaluating intertemporal and cross-generational policy effects.

Table 8: Stepwise mortality rates by age group (CDC, 2022[1]). Rates are per 100,000 population.

| Age Group (Years) | Mortality Rate (per 100,000) |
|---|---|
| $< 1$ | 560.0 |
| 1–4 | 28.0 |
| 5–14 | 15.3 |
| 15–24 | 79.5 |
| 25–34 | 163.4 |
| 35–44 | 255.4 |
| 45–54 | 453.3 |
| 55–64 | 992.1 |
| 65–74 | 1978.7 |
| 75–84 | 4708.2 |
| 85+ | 14389.6 |

### C.1.1   Markov Games

This MDP representation captures an individual's economic status and decision-making process, while ensuring compatibility with both the Ramsey model (featuring infinitely-lived agents) and the Overlapping Generations (OLG) model (featuring distinct life-cycle stages).

---

[3]https://www.federalreserve.gov/econres/scfindex.htm

Table 9: MDP Elements for Individual Agents in EconGym. Distributional statistics include wealth distribution, education distribution, wage rate, price level, lending rate, deposit rate.

| | Ramsey agents | OLG Agents |
|---|---|---|
| **Observation** | $o_t^i = (a_t^i, e_t^i)$ 
 Private: assets and education 
 Global: distributional statistics | $o_t^i = (a_t^i, e_t^i, \text{age}_t^i)$ 
 Private: assets, education, and **age** 
 Global: distributional statistics |
| **Action** | $a_t^i = (\alpha_t^i, \lambda_t^i, \theta_t^i)$ 
 Asset allocation, labor ratio, investment ratio 
 All actions available each period | $a_t^i = (\alpha_t^i, \lambda_t^i, \theta_t^i)$ 
 Same structure; age determines action validity 
 Old agents: $\lambda_t^i = 0$ (no labor) |
| **Reward** | $r_t^i = U(c_t^i, h_t^i)$ 
 CRRA utility over consumption and labor supply | $r_t^i = U(c_t^i, h_t^i)$ 
 Same form, but income includes pension if retired |

**Observation Space** At each time step $t$, an individual's observation consists of both *personal* and *global* components. The personal observation is defined as:

$$o_t^i = \left( a_t^i, e_t^i, \text{age}_t^i \right),$$

where:

- $a_t^i$ denotes the individual's asset holdings, including both savings and risky investments.
- $e_t^i$ represents the individual's education level, which influences wage income.
- $\text{age}_t^i$ is the individual's age, used only in the OLG model.

In addition, all individuals share access to a *global observation* vector summarizing population-level statistics. This includes aggregate indicators such as wealth distribution, education distribution, wage rate, price level, lending rate, deposit rate:

$$o_t^{\text{global}} = \left( \bar{a}^{\text{top10}}, \bar{e}^{\text{top10}}, \bar{a}^{\text{bot50}}, \bar{e}^{\text{bot50}}, W_{t-1}, P_{t-1}, r_{t-1}^l, r_{t-1}^d \right)$$

where each term represents the mean value within the corresponding subgroup. These global signals capture evolving socioeconomic structures and guide agents' strategic decisions in a shared environment.

**Action Space** At each time step $t$, the individual selects an action vector:

$$a_t^i = \left( \alpha_t^i, \lambda_t^i, \theta_t^i \right),$$

where:

- $\alpha_t^i \in [0, 1]$: allocation ratio indicating the share of disposable wealth allocated to financial planning (i.e., non-consumption).
- $\lambda_t^i \in [0, 1]$: labor effort ratio, representing the fraction of maximum working hours $h_{\max}$ in timestep $t$. Actual working hours are computed as $h_t^i = \lambda_t^i \cdot h_{\max}$. For retirees, $\lambda_t^i = 0$.
- $\theta_t^i \in [0, 1]$: investment ratio determining the fraction of allocated wealth invested in risky assets.

The action is subject to a budget constraint that governs the trade-off between consumption, saving, and labor income. Borrowing is restricted to ensure solvency, such that $a_{t+1}^i \geq a_{\min}$.

**Reward Function** The individual's objective is to maximize lifetime utility, with per-period reward given by:

$$R(s_t^i, a_t^i) = \frac{(c_t^i)^{1-\sigma}}{1-\sigma} - \frac{(h_t^i)^{1+\gamma}}{1+\gamma},$$

which reflects the trade-off between consumption utility and the disutility of labor.

## C.2 Government

In macroeconomic systems, the **government** represents a set of institutional agents tasked with managing fiscal stability, monetary policy, and social insurance. In EconGym, we implement three types of government agents: the **Fiscal Authority**, the **Central Bank**, and the **Pension Authority**. All government agents share the same global observation, but differ in their action spaces (i.e., macroeconomic policy) and optimization objectives (i.e., reward functions).

Table 10: Household Variables

| Variable | Meaning |
|---|---|
| $N_t$ | Total population size at time $t$ |
| $x_t$ | Share of young individuals in the population at time $t$ |
| $n_t^b$ | Birth rate at time $t$ |
| $n_t^d$ | Death rate at time $t$ |
| $age_t^i$ | Age of individual $i$ at time $t$ |
| $a_t^i$ | Total assets held by individual $i$ (savings + investment) |
| $s_t^i$ | Savings held by individual $i$ |
| $v_t^i$ | Risky investment held by individual $i$ |
| $c_t^i$ | Consumption of individual $i$ |
| $h_t^i$ | Labor supply (work effort) of individual $i$ |
| $e_t^i$ | Education level of individual $i$ (affecting productivity) |
| $W_t$ | Wage rate at time $t$ |
| $p_t$ | Price level at time $t$ |
| $r_t, \rho_t$ | Returns on savings and risky investments, respectively |
| $m_t^i$ | Disposable income of individual $i$ after tax (and pension contribution if young) |
| $\alpha_t^i$ | Allocation ratio: share of $m_t^i$ allocated to savings/investment |
| $\theta_t^i$ | Investment ratio: share of $a_{t+1}^i$ allocated to risky investment |
| $a_{t+1}^i$ | Next-period asset holdings after financial allocation |
| $f_t^y$ | Mandatory pension contribution for young individuals |
| $f_t^o$ | Pension benefit received by old individuals |
| $T_t(\cdot)$ | Tax function applied to income and capital |
| $\beta$ | Time discount factor |
| $\sigma$ | Coefficient of relative risk aversion |
| $\gamma$ | Inverse Frisch elasticity of labor supply |
| $o_t^i$ | Personal observation of individual $i$ at time $t$ |
| $o_t^{\text{global}}$ | Global observation vector shared by all agents |

**Fiscal Authority** The Fiscal Authority governs taxation and public spending to stimulate economic activity and maintain fiscal balance. It issues government bonds $B_t$, collects taxes via a function $T(\cdot)$, and allocates resources through public purchases $G_t$. Its operations are subject to the intertemporal budget constraint:

$$(1 + r_{t-1})B_t + G_t = B_{t+1} + T_t, \tag{2}$$

where $r_t$ is the nominal interest rate, $B_t$ is the outstanding government debt, and $T_t$ denotes total tax revenue collected at time $t$. In EconGym, we implement a nonlinear HSV tax function to model income and asset taxation:

$$T(i_t) = i_t - (1 - \tau)\frac{i_t^{1-\xi}}{1 - \xi}, \quad T^a(a_t) = a_t - \frac{1 - \tau_a}{1 - \xi_a}a_t^{1-\xi_a}, \tag{3}$$

where $i_t$ is taxable income and $a_t$ denotes asset holdings. Parameters $\tau, \tau_a$ control average tax rates, while $\xi, \xi_a$ determine tax curvature and progressivity. Alternatively, progressive schedules—such as the U.S. federal income tax—can also be specified.

**Central Bank** The Central Bank manages monetary policy to stabilize inflation and foster long-term growth. It adjusts the reserve requirement $\phi_t$ and the nominal interest rate $\iota_t$ to minimize deviations from policy targets:

$$\min_{\phi_t, \iota_t} \sum_t \left[ (\pi_t - \pi^*)^2 + \lambda_\pi (g_t - g^*)^2 \right], \tag{4}$$

where $\pi_t$ is the inflation rate, computed as:

$$\pi_t = \frac{P_t - P_{t-1}}{P_{t-1}},$$

and $g_t$ is the real GDP growth rate:

$$g_t = \frac{Y_t - Y_{t-1}}{Y_{t-1}}.$$

Here, $P_t$ denotes the aggregate price level (i.e., the consumer price index), and $Y_t$ denotes nominal GDP. The inflation target $\pi^*$ and growth target $g^*$ are exogenously set by national policy—typically around $\pi^* \approx 2\%$ and $g^* \approx 5\%$.

**Pension Authority** The Pension Authority manages intergenerational transfers through a national pension system. It collects contributions from working-age (young) households and pays benefits to retired (old) households. Its key policy parameters include the statutory retirement age $\text{age}^r$, the contribution rate $\tau_p$, and the benefit formula—jointly determining both fund sustainability and individual adequacy.

The pension fund evolves according to the following dynamics:

$$F_{t+1} = (1 + r_t^f)F_t + \sum_{i \in \mathcal{N}_t^y} P_t^y(i) - \sum_{i \in \mathcal{N}_t^o} P_t^o(i), \tag{5}$$

$$F_{t+1} \geq (1 + k)F_t, \tag{6}$$

where $F_t$ is the total fund value, $r_t^f$ is the pension investment return rate, and $k$ is the minimum required growth rate. The sets $\mathcal{N}_t^y$ and $\mathcal{N}_t^o$ denote young and old populations, respectively. Contributions are computed as:

$$P_t^y(i) = \tau_p \cdot E_t(i), \quad \text{with } E_t(i) = W_t \cdot e_t^i \cdot h_t^i,$$

where $W_t$ is the wage rate, $e_t^i$ is education level, and $h_t^i$ is labor effort.

**Dual-Component Pension Formula** In the Chinese pension system, monthly pension benefits consist of two components: a basic benefit and a personal account benefit. Specifically:

$$P_t^o(i) = P_t^{\text{basic}}(i) + P_t^{\text{personal}}(i), \tag{7}$$

$$P_t^{\text{basic}}(i) = \left( \frac{E_t^{\text{avg}} + E_t^{\text{ind}}(i)}{2} \right) \cdot T_i^p \cdot 0.01, \tag{8}$$

$$P_t^{\text{personal}}(i) = \frac{A_t^{\text{personal}}(i)}{M}, \tag{9}$$

where $E_t^{\text{avg}}$ is the national average wage, $E_t^{\text{ind}}(i)$ is the individual's average wage over contribution years $T_i^p$, and $A_t^{\text{personal}}(i)$ is the accumulated personal account balance:

$$A_t^{\text{personal}}(i) = \sum_{s=0}^{T_i^p} \left( P_s^y(i) \cdot (1 + r_s^f)^{t-s} \right). \tag{10}$$

The annuity divisor $M$ depends on the individual's retirement age, as listed in Table 11.

The government sets the retirement age and contribution parameters to ensure both the solvency of the pension system and fairness across generations.

### C.2.1 Markov Games

In EconGym, each government agent—**Fiscal Authority**, **Central Bank**, and **Pension Authority**—is modeled as an independent decision-maker operating under a Markov Decision Process (MDP). These agents share a common macroeconomic observation but differ in action spaces and optimization objectives.

**Fiscal Authority** *Observation Space:*

$$o_t^{\text{fiscal}} = \{\mathcal{A}_t, \mathcal{E}_{t-1}, W_{t-1}, P_{t-1}, r_{t-1}^l, r_{t-1}^d, B_{t-1}\},$$

where:

- $\mathcal{A}_t$: wealth distribution, represented by the average wealth of the top 10% richest individuals and the bottom 50% poorest individuals, $\bar{a}^{\text{top10}}, \bar{a}^{\text{bot50}}$.
- $\mathcal{E}_{t-1}$: education distribution, e.g., the average education level of the top 10% richest individuals and the bottom 50% poorest individuals, $\bar{e}^{\text{top10}}, \bar{e}^{\text{bot50}}$.

Table 11: Annuity Factor $M$ by Retirement Age in China's Pension System

| Retirement Age | Annuity Factor $M$ | Retirement Age | Annuity Factor $M$ |
|---|---|---|---|
| 40 | 233 | 56 | 164 |
| 41 | 230 | 57 | 158 |
| 42 | 226 | 58 | 152 |
| 43 | 223 | 59 | 145 |
| 44 | 220 | 60 | 139 |
| 45 | 216 | 61 | 132 |
| 46 | 212 | 62 | 125 |
| 47 | 208 | 63 | 117 |
| 48 | 204 | 64 | 109 |
| 49 | 199 | 65 | 101 |
| 50 | 195 | 66 | 93 |
| 51 | 190 | 67 | 84 |
| 52 | 185 | 68 | 75 |
| 53 | 180 | 69 | 65 |
| 54 | 175 | 70 | 56 |
| 55 | 170 | | |

Table 12: MDP Elements for Government Agents in EconGym

| | Fiscal Authority | Central Bank | Pension Authority |
|---|---|---|---|
| Observation | $o_t^{\text{fiscal}} = \{\mathcal{A}_t, \mathcal{E}_{t-1}, W_{t-1}, P_{t-1}, r^l_{t-1}, r^d_{t-1}, B_{t-1}\}$ Wealth distribution, education distribution, wage rate, price level, lending rate, deposit rate, debt. | $o_t^{\text{cb}} = \{\mathcal{A}_t, \mathcal{E}_{t-1}, W_{t-1}, P_{t-1}, r^l_{t-1}, r^d_{t-1}, \pi_{t-1}, g_{t-1}\}$ Wealth distribution, education distribution, wage rate, price level, lending rate, deposit rate, inflation rate, growth rate. | $o_t^{\text{pension}} = \{F_{t-1}, N_t, N_t^{old}, \text{age}^r_{t-1}, \tau^p_{t-1}, B_{t-1}, Y_{t-1}\}$ Pension fund, current population, old individuals' number, last retirement age, last contribution rate, debt, GDP |
| Action | $a_t^{\text{fiscal}} = \{\boldsymbol{\tau}, G_t\}$ Tax rates and gov. spending | $a_t^{\text{cb}} = \{\phi_t, \iota_t\}$ Reserve ratio and benchmark rate | $a_t^{\text{pension}} = \{\text{age}^r, \tau_p, k\}$ Retirement age, contribution rate, target growth |
| Reward | GDP growth, inequality reduction, or welfare maximization | Inflation/GDP stabilization | Pension sustainability via fund growth $\left(\frac{Y_t}{Y_{t-1}} - 1\right)$ |

- $W_{t-1}$: wage rate in the economy.
- $P_{t-1}$: price level of goods and services in the economy.
- $r^l_{t-1}$: lending rate charged by banks for loans.
- $r^d_{t-1}$: deposit rate paid by banks on deposits.
- $B_{t-1}$: public government debt at time $t-1$.

*Action Space:*

$$a_t^{\text{fiscal}} = \{\boldsymbol{\tau}, G_t\},$$

where:

- $\boldsymbol{\tau}$: income and asset tax parameters (e.g., $\tau, \xi, \tau_a, \xi_a$).
- $G_t$: government spending as a fraction of GDP.

*Reward Function:* The Fiscal Authority may pursue one of several policy goals, such as:

$$r_t^{\text{fiscal}} = \left(\frac{Y_t}{Y_{t-1}}\right) - 1 \quad \text{(maximize GDP growth)},$$

$$r_t^{\text{fiscal}} = 1 - \text{Gini}(\mathcal{I}_t) \quad \text{(minimize income inequality)},$$

$$r_t^{\text{fiscal}} = \sum_{i=1}^{N} u(c_t^i, h_t^i) \quad \text{(maximize aggregate welfare)}.$$

**Central Bank** *Observation Space:*

$$o_t^{\text{cb}} = \{\mathcal{A}_t, \mathcal{E}_{t-1}, W_{t-1}, P_{t-1}, r^l_{t-1}, r^d_{t-1}, \pi_{t-1}, g_{t-1}\},$$

where:

- $\mathcal{A}_t$: wealth distribution, the overall wealth dispersion in the economy.
- $\mathcal{E}_{t-1}$: education distribution, representing the average education level across different income groups.
- $W_{t-1}$: wage rate in the economy.
- $P_{t-1}$: price level of goods and services in the economy.
- $r^l_{t-1}$: lending rate charged by banks for loans.
- $r^d_{t-1}$: deposit rate paid by banks on deposits.
- $\pi_{t-1}$: inflation rate, the rate at which the general price level has increased over the past period.
- $g_{t-1}$: growth rate, the rate at which real GDP has expanded in the economy.

*Action Space:*
$$a^{\text{cb}}_t = \{\phi_t, \iota_t\},$$
where:

- $\phi_t$: reserve requirement ratio, the fraction of bank deposits that must be kept in reserve.
- $\iota_t$: nominal interest rate, the benchmark interest rate set by the central bank.

*Reward Function:* The Central Bank aims to jointly stabilize inflation and sustain growth via:
$$r^{\text{cb}}_t = \exp\left(-\left[(\pi_t - \pi^*)^2 + \lambda_\pi(g_t - g^*)^2\right]\right),$$
where:

- $\pi_t = \frac{P_t - P_{t-1}}{P_{t-1}}$: current inflation.
- $g_t = \frac{Y_t - Y_{t-1}}{Y_{t-1}}$: real GDP growth.
- $\pi^*$: target inflation (e.g., 2%), $g^*$: target growth (e.g., 5%).
- $\lambda_\pi$: trade-off weight between inflation stability and growth.

**Pension Authority**   *Observation Space:*
$$o^{\text{pension}}_t = \{F_{t-1}, N_t, N^{old}_t, \text{age}^r_{t-1}, \tau^p_{t-1}, B_{t-1}, Y_{t-1}\},$$
where:

- $F_{t-1}$: pension fund in the pension system.
- $N_t$: current population in the economy.
- $N^{old}_t$: number of old individuals, those who are above the retirement age.
- $\text{age}^r_{t-1}$: last retirement age, the age at which individuals were eligible for retirement.
- $\tau^p_{t-1}$: last contribution rate, the percentage of income contributed to the pension system.
- $B_{t-1}$: public government debt at time $t-1$.
- $Y_{t-1}$: GDP, the total economic output of the economy.

*Action Space:*
$$a^{\text{pension}}_t = \{\text{age}^r, \tau_p, k\},$$
where:

- $\text{age}^r$: statutory retirement age, the legal age at which individuals can retire.
- $\tau_p$: pension contribution rate, the percentage of income contributed to the pension fund.
- $k$: required pension fund growth rate, the desired rate at which the pension fund grows.

*Reward Function:* The Pension Authority focuses on supporting long-term sustainability through growth:
$$r^{\text{pension}}_t = \left(\frac{Y_t}{Y_{t-1}}\right) - 1.$$

This modular MDP design allows each government agent to optimize policies independently while interacting with a shared economic environment.

Table 13: Government Variables

| Symbol | Definition |
|---|---|
| $B_t$ | Government bond holdings at time $t$ |
| $T(i_t), T^a(a_t)$ | Income and asset tax functions |
| $\tau, \tau_a$ | Average tax rate for income and assets |
| $\xi, \xi_a$ | Curvature parameters for nonlinear tax function |
| $G_t$ | Government spending on goods and services |
| $r_t$ | Risk-free interest rate on government debt |
| $Y_t$ | Nominal Gross Domestic Product (GDP) at time $t$ |
| $P_t$ | Aggregate price level (CPI) at time $t$ |
| $\pi_t$ | Inflation rate: $\pi_t = \frac{P_t - P_{t-1}}{P_{t-1}}$ |
| $g_t$ | GDP growth rate: $g_t = \frac{Y_t - Y_{t-1}}{Y_{t-1}}$ |
| $\pi^*$ | Target inflation rate set by monetary authority |
| $g^*$ | Target GDP growth rate |
| $\phi_t$ | Reserve requirement ratio set by central bank |
| $\iota_t$ | Nominal interest rate set by central bank |
| $\lambda_\pi$ | Trade-off weight between inflation and growth objectives |
| $F_t$ | Total value of the national pension fund |
| $r_t^f$ | Investment return rate of the pension fund |
| $k$ | Minimum required pension fund growth rate |
| $\tau_p$ | Pension contribution rate paid by working individuals |
| $P_t^y(i)$ | Pension contribution from individual $i$ at time $t$ |
| $P_t^o(i)$ | Pension benefit received by retired individual $i$ |
| $A_t^{\text{personal}}(i)$ | Accumulated balance of personal pension account |
| $M$ | Annuity divisor based on retirement age |
| $\text{age}^r$ | Statutory retirement age |
| $T_i^p$ | Total years of contribution for individual $i$ |
| $\mathcal{I}_t$ | Income distribution across the population at time $t$ |
| $u(c_t^i, h_t^i)$ | Instantaneous utility from consumption and labor of individual $i$ |

## C.3 Bank

EconGym also models financial intermediaries (hereafter referred to as **banks**), which play a critical role in channeling funds, managing liquidity, and transmitting monetary policy. We consider two types of bank agents: the **Non-Profit Financial Platform** and the **Commercial Bank**. Both institutions collect household deposits and allocate them across productive capital and government bonds, but they differ significantly in their decision-making behavior and regulatory constraints.

**Non-Profit Financial Platform**    The non-profit financial platform provides essential deposit and lending services without any profit-maximization motive. Unlike commercial banks, it does not control interest rates or make strategic decisions—hence, it is not modeled as a Markov Decision Process (MDP) agent. Instead, the platform passively intermediates household savings, allocating them across productive capital for firms and government bonds. Let $A_t = \sum_{i=1}^{N} s_t^i$ denote the total household savings at time $t$, where $s_t^i$ is individual $i$'s deposit. Let $K_t$ represent the total capital lent to firms, and $B_t$ the platform's holdings of government bonds.

The platform's budget evolves according to:

$$K_{t+1} + B_{t+1} - A_{t+1} = (R_t + 1 - \delta)K_t + (1 + r_{t-1})(B_t - A_t), \tag{11}$$

where:

- $R_t$: rental rate of capital at time $t$,
- $\delta$: capital depreciation rate,
- $r_t$: nominal return on government bonds.

To maintain equilibrium between capital and bond markets, a no-arbitrage condition is imposed:

$$R_{t+1} = r_t + \delta. \tag{12}$$

This ensures that both capital investments and bond purchases offer equivalent returns in expectation, preserving incentive neutrality in the platform's passive allocation mechanism.

**Commercial Bank** In contrast to the passive financial platform, the **commercial bank** actively manages interest rates to maximize long-run profitability. It determines the deposit rate $r_t^d$ and the lending rate $r_t^l$, which apply uniformly across all depositors and borrowers. These decisions are made under regulatory constraints imposed by the central bank, including a reserve requirement ratio $\phi \in [0, 1]$ and bounds on permissible interest spreads.

At each time step, the commercial bank receives household deposits $A_t$ and allocates them between two asset classes: productive capital loans $K_t$ and government bonds $B_t$. Its profit in period $t$ is defined as:

$$\Pi_t = r_{t-1}^l (K_{t-1} + B_{t-1}) - r_{t-1}^d A_t, \tag{13}$$

capturing the margin between returns on lending and bonds versus interest paid on deposits. The cumulative objective is to maximize long-term profit:

$$\max_{r^d, r^l} \sum_{t=0}^{T} \Pi_t = \sum_{t=0}^{T} \left[ r_t^l (K_{t+1} + B_{t+1}) - r_t^d A_{t+1} \right]. \tag{14}$$

**Constraints of central bank:** The bank's asset allocation must respect a liquidity constraint that reflects the central bank's reserve requirement:

$$K_t + B_t \leq (1 - \phi) A_t, \tag{15}$$

ensuring that a fraction $\phi A_t$ of deposits is held in reserve and cannot be loaned or invested.

Additionally, the central bank enforces bounds around the benchmark interest rate $\iota_t$ to constrain commercial bank behavior:

$$\iota_t - 0.01 \leq r_t^d \leq \iota_t, \tag{16}$$

$$\iota_t + 0.01 \leq r_t^l \leq \iota_t + 0.03. \tag{17}$$

These bounds ensure a stable interest rate corridor, maintaining incentives for financial intermediation while embedding monetary control.

### C.3.1 Markov Games

Table 14: MDP Elements for Bank Agents in EconGym

| | Non-Profit Platform | Commercial Bank |
|---|---|---|
| **Observation** | — | $o_t^{\text{bank}} = \{\iota_t, \phi_t, r_{t-1}^l, r_{t-1}^d, loan, F_{t-1}\}$ |
| | — | Benchmark rate, reserve ratio, last lending rate, last deposit rate, loans, pension fund. |
| **Action** | Static allocation rule | $a_t^{\text{bank}} = \{r_t^d, r_t^l\}$ |
| | No control over rates | Deposit and lending rate decisions |
| **Reward** | — | $a_t^{\text{bank}} = r_t^l (K_{t+1} + B_{t+1}) - r_t^d A_{t+1}$ |
| | Passive agent | Interest margin under reserve constraint |

**Observation Space** At time $t$, the commercial bank observes:

$$o_t^{\text{bank}} = \{\iota_t, \phi_t, r_{t-1}^l, r_{t-1}^d, loan, F_{t-1}\},$$

including the central bank's benchmark rate, reserve ratio, last lending rate, last deposit rate, loans, pension fund.

**Action Space** The commercial bank decides:

$$a_t^{\text{bank}} = \{r_t^d, r_t^l\},$$

subject to regulatory interest bounds and the reserve constraint.

Table 15: Bank Variables

| Symbol | Definition |
|---|---|
| $A_t$ | Net household deposits at time $t$ (total deposits minus outstanding loans) |
| $K_t$ | Productive capital lent to firms at time $t$ |
| $B_t$ | Government bonds held by the bank at time $t$ |
| $s_t^i$ | Individual $i$'s savings (deposit) at time $t$ |
| $R_t$ | Rental rate of capital at time $t$ |
| $r_t$ | Nominal return on government bonds at time $t$ |
| $\delta$ | Capital depreciation rate |
| $\Pi_t$ | Commercial bank's profit at time $t$ |
| $r_t^d$ | Deposit interest rate set by the commercial bank at time $t$ |
| $r_t^l$ | Lending interest rate set by the commercial bank at time $t$ |
| $\iota_t$ | Central bank's benchmark interest rate at time $t$ |
| $\phi$ | Reserve requirement ratio imposed by the central bank |
| $o_t^{\text{bank}}$ | Bank observation at time $t$ |
| $a_t^{\text{bank}}$ | Bank action at time $t$ (e.g., $r_t^d, r_t^l$) |
| $r_t^{\text{bank}}$ | Bank reward (net interest profit) at time $t$ |

**Reward Function**  The reward is the profit from interest margin:

$$r_t^{\text{bank}} = \Pi_t = r_t^l(K_{t+1} + B_{t+1}) - r_t^d A_{t+1}.$$

This formulation enables the commercial bank to adaptively manage its pricing and investment decisions in response to macroeconomic signals, while adhering to monetary policy and liquidity regulations.

## C.4  Firms

In economic theory, a **firm** is a production unit that transforms capital and labor inputs into output, typically aiming to maximize profit subject to technological and market constraints [26, 44]. Firm behavior plays a critical role in shaping prices, income distribution, and overall macroeconomic performance [5]. To capture a wide range of market environments, EconGym supports four canonical firm types: **Perfect Competition**, **Monopoly**, **Oligopoly**, and **Monopolistic Competition**. While all firms share a common production foundation, they differ in pricing power, market structure, and strategic decision-making.

Each firm produces output $Y_t$ using a Cobb-Douglas production function:

$$Y_t = Z_t K_t^\alpha L_t^{1-\alpha}, \tag{18}$$

where $K_t$ and $L_t$ denote the firm's capital and labor inputs at time $t$, $\alpha \in (0, 1)$ is the capital elasticity of output, and $Z_t$ is total factor productivity. The evolution of productivity is governed by a stochastic process:

$$\log(Z_t) = \log(Z_{t-1}) + \sigma_z \epsilon_t, \quad \epsilon_t \sim \mathcal{N}(0, 1), \tag{19}$$

where $\sigma_z$ captures the volatility of idiosyncratic productivity shocks.

Firm profit is computed as:

$$\Pi_t = p_t Y_t - W_t L_t - R_t K_t, \tag{20}$$

with $p_t$ denoting the price of goods, $W_t$ the wage rate, and $R_t$ the rental rate of capital.

**Perfect Competition**  Firms in a perfectly competitive market operate under the assumption of *no market power*: they take prices, wages, and capital rental rates as given. Products are homogeneous, and free entry drives long-run profits to zero.

Since firms are price takers, the output price $p_t$ is determined by market-clearing conditions:

$$p_t Y_t = p_t C_t + G_t + I_t, \tag{21}$$

ensuring that total supply equals total demand. Factor prices are derived from marginal product conditions. The equilibrium wage rate is:

$$W_t = (1 - \alpha)p_t Z_t \left(\frac{K_t}{L_t}\right)^\alpha, \tag{22}$$

and the rental rate of capital is:

$$R_t = \alpha p_t Z_t \left(\frac{K_t}{L_t}\right)^{\alpha - 1}. \tag{23}$$

Substituting Eqs. (22) and (23) into the profit equation (Eq. (20)) leads to:

$$\Pi_t = 0.$$

This confirms that firms in perfect competition earn zero economic profits in equilibrium—a key result in classical price theory.

Perfectly competitive firms are not modeled as decision-making agents within EconGym. Since prices and factor returns are externally determined and all firms behave identically, their production responds passively to market signals without strategic optimization. As such, they do not constitute a Markov Decision Process (MDP) agent.

**Monopoly**    A monopoly describes a market with a single firm that faces no competition. As the sole producer, the firm has the ability to set both the output price $p_t$ and the wage rate $W_t$, rather than taking them as given. However, this pricing power is limited by household demand and broader macroeconomic conditions.

The firm's output follows the standard Cobb-Douglas form (Eq. 18), and profit is defined by:

$$\Pi_t = p_t Y_t - W_t L_t - R_t K_t. \tag{24}$$

where $R_t$ is the rental rate of capital. Unlike firms in perfect competition, the monopolist chooses both $p_t$ and $W_t$ to maximize profit.

Demand for the monopolist's goods and labor is shaped by household and government budget constraints. Households follow an intertemporal consumption-saving decision:

$$p_t C_t + A_{t+1} = (1 + r_t)A_t + W_t L_t - T_t, \tag{25}$$

where $A_t$ is the total household asset and $T_t$ is the tax burden. The government maintains fiscal balance:

$$B_{t+1} + T_t = G_t + (1 + r_t)B_t, \tag{26}$$

where $B_t$ denotes government debt and $G_t$ is government spending.

**Oligopoly**    An oligopoly features a small number of competing firms that each possess market power. Firms independently set prices and wages, anticipating household responses and rivals' behavior.

We model $N_f$ symmetric firms indexed by $j \in \{1, \ldots, N_f\}$. Each firm produces:

$$y_t^j = Z_t^j (K_t^j)^\alpha (L_t^j)^{1-\alpha}, \tag{27}$$

where productivity $Z_t^j$ follows a log-normal stochastic process. The firm sets a product price $p_t^j$ and wage $W_t^j$, and earns profit:

$$\Pi_t^j = p_t^j y_t^j - W_t^j L_t^j - R_t K_t^j. \tag{28}$$

Revenue is generated from private consumption, government purchases, and reinvestment:

$$p_t^j y_t^j = p_t^j \left(\sum_{i=1}^N c_t^{ij}\right) + G_t^j + I_t^j. \tag{29}$$

Capital evolves over time:

$$K_{t+1}^j = I_t^j + (1 - \delta)K_t^j. \tag{30}$$

Households observe all firm prices $\{p_t^j\}$ and wages $\{W_t^j\}$, and choose a single firm for consumption and labor. This induces firm-specific demand via discrete choice and budget allocation. As a result, each firm's output and labor demand depend on its own pricing strategy and its relative attractiveness to households.

**Monopolistic Competition** Under monopolistic competition, a large number of firms offer differentiated products. Each firm has limited pricing power but faces zero long-run profits due to free entry.

Firm $j$ solves:

$$\max_{p_t^j, W_t^j} \Pi_t^j = p_t^j y_t^j - W_t^j L_t^j - R_t K_t^j. \tag{31}$$

with production:

$$y_t^j = Z_t^j (K_t^j)^\alpha (L_t^j)^{1-\alpha}. \tag{32}$$

First-order conditions yield the optimal capital-labor ratio:

$$\frac{K_t^j}{L_t^j} = \frac{\alpha W_t^j}{(1-\alpha)R_t}. \tag{33}$$

Labor demand is:

$$L_t^j = \frac{y_t^j}{Z_t^j} \left( \frac{(1-\alpha)R_t}{\alpha W_t^j} \right)^\alpha. \tag{34}$$

Assuming market clearing:

$$L_t^j = \sum_{i=1}^{N} h_t^{ij}. \tag{35}$$

Marginal cost is constant:

$$\mathrm{MC}_t^j = \frac{W_t^{1-\alpha} R_t^\alpha}{Z_t^j \alpha^\alpha (1-\alpha)^{1-\alpha}}. \tag{36}$$

Price is set using a markup rule:

$$p_t^j = \frac{\epsilon}{\epsilon - 1} \cdot \mathrm{MC}_t^j. \tag{37}$$

Aggregate prices are determined via CES aggregation:

$$P_t = \left( \sum_{j=1}^{N_f} (p_t^j)^{1-\epsilon} \right)^{\frac{1}{1-\epsilon}}, \tag{38}$$

and household consumption is given by:

$$c_t^i = \left( \sum_{j=1}^{N_f} (c_t^{ij})^{\frac{\epsilon-1}{\epsilon}} \right)^{\frac{\epsilon}{\epsilon-1}}. \tag{39}$$

Each firm optimizes profit under a stable demand system shaped by its pricing decision and product differentiation.

### C.4.1 Markov Games

**Monopoly Firm**

*Action Space:*

$$a_t^{\mathrm{mono}} = \{p_t, W_t\},$$

- $p_t$: product price
- $W_t$: wage offered to workers

*Observation:*

$$o_t^{\mathrm{mono}} = \{K_t, Z_t, r_{t-1}^l\}$$

*Reward Function:*

$$r_t^{\mathrm{mono}} = p_t Y_t - W_t L_t - R_t K_t$$

Table 16: MDP Elements for Firm Agents in EconGym

| | Perfect Competition | Monopoly | Oligopoly | Monopolistic Competition |
|---|---|---|---|---|
| **Observation** | — | $o_t^{\text{firm}} = \{K_t, Z_t, r_{t-1}^l\}$ Capital, labor, productivity, previous price and wage | | |
| **Action** | — | $a_t^{\text{firm}} = \{p_t, W_t\}$ Product price and wage setting | | |
| **Reward** | — | $r_t^{\text{firm}} = p_t Y_t - W_t L_t - R_t K_t$ Profits = Revenue minus wage and capital costs | | |
| **Price &Wage Setting** | ✗ (market-clearing) | ✓ | ✓ | ✓ |
| **Product Type** | Homogeneous | Single good | Differentiated (per firm) | Differentiated (CES aggregate) |
| **Firm Number** | Many | One | Few ($N_f$) | Many ($N_f$) |
| **Strategic Interaction** | None | With households | With firms and households | With households (via CES demand) |
| **Long-run Profits** | Zero | Non-zero | Firm-dependent | Zero (free entry) |

**Oligopoly Firm $j$**

*Action Space:*

$$a_t^{\text{olig}} = \{p_t^j, W_t^j\}$$

- $p_t^j$: firm $j$'s product price
- $W_t^j$: wage set by firm $j$

*Observation:*

$$o_t^{\text{olig}} = \{K_t^j, Z_t^j, r_{t-1}^l\}$$

*Reward Function:*

$$r_t^{\text{olig}} = p_t^j y_t^j - W_t^j L_t^j - R_t K_t^j$$

**Monopolistic Competition Firm**

*Action Space:*

$$a_t^{\text{mono-comp}} = \{p_t^j, W_t^j\}$$

*Observation:*

$$o_t^{\text{mono-comp}} = \{K_t^j, Z_t^j, r_{t-1}^l\}$$

*Reward Function:*

$$r_t^{\text{mono-comp}} = p_t^j y_t^j - W_t^j L_t^j - R_t K_t^j$$

Each firm observes its own productivity and factor inputs, sets prices and wages, and receives profit as its reward signal. Firms in monopoly and oligopoly settings face strategic competition or feedback from endogenous demand, whereas firms in monopolistic competition operate under fixed CES preferences and long-run zero-profit equilibrium.

# D  Validation against Economic Benchmarks and Peer Platforms

To ensure the credibility of EconGym's modeling and simulation design, we provide additional validation from two complementary perspectives: (1) consistency with established economic benchmarks, and (2) computational comparison with representative peer platforms.

**Validation against established economic benchmarks.**  We validate EconGym against the canonical benchmark in the DSGE family, the Aiyagari model [2], which remains the foundation of heterogeneous-agent macroeconomic analysis [17]. Using standard calibration parameters (capital share $\alpha = 0.36$, depreciation $\delta = 0.06$, productivity $Z = 1$), we simulate the optimal-tax scenario in EconGym and report steady-state inequality metrics. As shown in Table 18, EconGym reproduces both income and wealth inequality ranges consistent with the Aiyagari benchmark, confirming alignment with widely accepted macroeconomic regularities.

Table 17: Firm Variables

| Symbol | Definition |
|---|---|
| $K_t, K_t^j$ | Capital stock used by the firm (aggregate or firm $j$) at time $t$ |
| $L_t, L_t^j$ | Labor employed by the firm (aggregate or firm $j$) at time $t$ |
| $Z_t, Z_t^j$ | Total factor productivity for the firm at time $t$ |
| $Y_t, y_t^j$ | Output produced by the firm (aggregate or firm $j$) at time $t$ |
| $I_t^j$ | Capital investment by firm $j$ at time $t$ |
| $p_t, p_t^j$ | Product price set by the firm (aggregate or firm $j$) at time $t$ |
| $W_t, W_t^j$ | Wage offered to workers by the firm at time $t$ |
| $R_t$ | Rental rate of capital at time $t$ |
| $P_t$ | Aggregate price index (CES-based) at time $t$ |
| $\epsilon$ | Elasticity of substitution in CES demand |
| $\mathrm{MC}_t^j$ | Marginal cost of firm $j$ at time $t$ |
| $\mathrm{TC}_t^j$ | Total cost of firm $j$ at time $t$ |
| $\Pi_t, \Pi_t^j$ | Profit of the firm (aggregate or firm $j$) at time $t$ |
| $a_t^{\mathrm{mono}}, a_t^{\mathrm{olig}}, a_t^{\mathrm{mono\text{-}comp}}$ | Firm action (e.g., price and wage decisions) at time $t$ |
| $s_t^{\mathrm{mono}}, s_t^{\mathrm{olig}}, s_t^{\mathrm{mono\text{-}comp}}$ | Firm state observation at time $t$ |
| $p_{t-1}, p_{t-1}^j$ | Lagged product price from previous timestep |
| $W_{t-1}, W_{t-1}^j$ | Lagged wage rate from previous timestep |
| $r_t^{\mathrm{mono}}, r_t^{\mathrm{olig}}, r_t^{\mathrm{mono\text{-}comp}}$ | Reward function (firm profit) at time $t$ |
| $\alpha$ | Output elasticity of capital in the production function |
| $\delta$ | Depreciation rate of capital |
| $\sigma_z$ | Volatility of productivity shock in log-normal process |

Table 18: Validation against the Aiyagari benchmark model.

| Model | Income Gini | Wealth Gini |
|---|---|---|
| Aiyagari model | 0.30–0.40 | $\approx 0.80$ |
| **EconGym (ours)** | $0.314 \pm 0.081$ | $0.728 \pm 0.010$ |

**Comparison with peer simulation platforms.** We further compare EconGym with representative AI-based economic simulators—the AI Economist [47] and TaxAI [27]. Although these platforms differ in modeling assumptions and task scopes, we follow standard benchmarking practice (e.g., OpenAI Gym) by comparing *environment sampling efficiency*, a key factor determining algorithmic scalability. As summarized in Table 19, EconGym attains substantially lower per-step simulation time while supporting larger agent populations and richer task diversity, enabling efficient large-scale policy learning and cross-method evaluation.

Table 19: Comparison of environment sampling efficiency.

| Platform | Agents | Sample Time (ms/step) |
|---|---|---|
| AI Economist [47] | 4 | 4.03 |
| TaxAI [27] | 100 | 2.073 |
| **EconGym (ours)** | 100 | **0.665** |

These results demonstrate that EconGym not only reproduces classical macroeconomic patterns but also achieves state-of-the-art computational scalability, providing a robust foundation for benchmarking AI-driven economic research.

# E    Illustrative LLM Outputs in the Pension Policy Scenario

To complement the quantitative analysis in Section 4.2, we present representative reasoning traces of the large language model (LLM) acting as a pension authority in the aging–pension scenario. These

examples, generated by the model `DeepSeek-V3-0324`, demonstrate its chain-of-thought (CoT) reasoning process and the ability to make interpretable, adaptive policy decisions.

---

**Prompt Template for Policy Model**

*System Prompt:* "You are a pension authority agent responsible for maintaining long-term economic sustainability. Given the current macroeconomic indicators (GDP growth, inflation, pension burden, and Gini index), propose adjustments to the retirement age and contribution rate. Explain your reasoning step by step, ensuring that your decisions balance individual welfare and macroeconomic stability."

---

**1. Understanding Causation: Explaining Policy Adjustments.**  The LLM justifies its policy decisions by clearly explaining the cause-and-effect relationships behind each adjustment. For example, when addressing the need for pension system sustainability, the LLM ties the increase in retirement age and contribution rate to specific economic indicators:

> *"To address this, a modest increase in the retirement age (from 62.0 to 62.5) is proposed to slightly extend working years without overly burdening workers, thereby sustaining the pension system and supporting GDP growth."*

This highlights the LLM's ability to reason through economic conditions and explain its decisions accordingly.

**2. Adapting to Economic Conditions.**  The LLM demonstrates a keen sensitivity to system prompts such as inflation and GDP growth. When faced with high inflation, the LLM adjusts the contribution rate while ensuring that disposable income is not significantly impacted:

> *"The contribution rate is increased from 0.15 to 0.16 to improve pension fund sustainability, as the current fund ratio is 0.0, indicating underfunding. This adjustment is cautious to avoid significantly reducing disposable income and consumption, given the high inflation (18.104%)."*

This shows how the LLM tailors its policy decisions to accommodate economic pressures, balancing long-term sustainability with immediate impacts.

**3. Evolving Decisions Over Time.**  The LLM adjusts its decisions as the economic environment evolves. For example, with **GDP growth** at 2.064%, it incrementally raises the retirement age to balance pension sustainability while accommodating a growing labor force:

> *"The retirement age is increased from 62.5 to 63.0 to address the high pension burden (21.6% of GDP) and sustain the pension system without overly burdening workers, given the relatively high GDP growth rate (2.064%) and mean household income (155,405.89)."*

This demonstrates how the LLM accounts for economic changes and revises policies in response to shifting conditions, emphasizing its ability to adjust dynamically.

**4. Addressing Wealth Inequality.**  The LLM also incorporates **wealth inequality** considerations into its decisions. In response to a high Gini index, the LLM adjusts both the retirement age and contribution rate to reduce inequality while maintaining economic stability:

> *"The retirement age is increased slightly from 63.0 to 64.0 to help sustain the pension system without overly burdening workers, given the high wealth Gini index (0.730) and projected pension burden (-0.156). This adjustment balances the need to maintain GDP growth (0.179) while minimizing work burden."*

This case illustrates how the LLM incorporates redistributive policy goals, making it a powerful tool for addressing socio-economic issues within the simulation.

**5. Ensuring Long-Term Sustainability.** The LLM carefully balances **short-term needs** with **long-term sustainability goals**. In response to a projected pension burden, it gradually increases the retirement age while moderating the contribution rate to ensure the system's viability:

> *"The decision to increase the retirement age to 67.0 (from 66.5) is based on the need to sustain the pension system given the high wealth Gini index (0.638) and projected pension burden (-0.452). A modest increase in retirement age helps balance the work burden while ensuring pension sustainability."*

These examples demonstrate the LLM's capacity for interpretable, data-informed reasoning across evolving macroeconomic conditions.

