# OpenReview forum: "EconGym: A Scalable AI Testbed with Diverse Economic Tasks"
_NeurIPS.cc/2025/Datasets_and_Benchmarks_Track — NeurIPS 2025 Datasets and Benchmarks Track poster_

### Official Review · Reviewer_WNuG · 2025-06-04

**Rating:** 5
**Confidence:** 4

**Summary:**

This paper presents EconGym, a multiagent simulator for economic activities focusing on domains such as taxes, interest rates, pension. Different tasks are composed of a shared set of individual agent types/roles with potentially distinct observation and action spaces. Agents can be driven by AI policies or economic models. Experiments demonstrate the flexibility of the environments, interesting economic insights, and benefits of combining economic models with AI agents.

**Dataset Code Accessibility:**

Yes

**Dataset Code Comments:**

I reviewed the anonymous Github site. Documentations in terms of how to design experiments for economic questions are comprehensive.

**Ethical Considerations:**

No, there are no or only very minor ethics concerns

**Final Justification:**

I believe that all reviewers agree that this paper is novel and the contributions are highly valuable and relevant.

**Limitations Weaknesses:**

* It's not obvious to me that there is any vital weakness in this paper, other than that the project is in development and a crucial step towards AI for real world economic applications.
* I am curious whether and how this platform could be used for incentive design, for example at the level of designing agent rewards or other incentive mechanisms.

**Strengths Contributions:**

* High quality economic simulators are highly valuable for economic decision making and policy design. EconGym provides more diversity, flexibility, compositionally, and grounding in economic theory than any prior simulators.

---

> ### Author Rebuttal · Authors · 2025-07-30
>
> **Dear Reviewer WNuG,**
>
> Thank you for your thoughtful review and positive comments. We sincerely appreciate your recognition of **EconGym**: “High quality economic simulators” and “EconGym provides more diversity, flexibility, compositionally, and grounding in economic theory than any prior simulators.”
>
> In rebuttal, we aim to address your concerns as thoroughly as possible and demonstrate our continuous efforts to advance the development and refinement of EconGym:
>
> - **Enhanced sampling efficiency of EconGym**, now supporting dynamic interactions with up to **100,000 individuals** within a tolerable sampling time. This improvement increases efficiency by **at least 10 times** compared to previous versions.
> - **Demonstrated how EconGym provides an experimental platform** for studying **incentive design** and evaluating its effectiveness.
> - **Listed tasks in EconGym** designed for studying incentive design, showcasing practical examples for research.
>
> **Below, we provide detailed responses to each of your questions to address every concern and ensure the clarity of our work.**
>
> ---
>
>
>
> > **W1: "It's not obvious to me that there is any vital weakness in this paper, other than that the project is in development and a crucial step towards AI for real world economic applications."**
>
> **Answer:** Thank you for your support and recognition of our EconGym project. **We are committed to continuously improving its development** and look forward to its contribution to AI-driven real-world economic applications.
>
> Recently, **we have further optimized the environment sampling efficiency of EconGym** by refining the computation of agent economic variables. Below is the latest statistical data for the **sampling efficiency (ms/step) across 25 tasks**. This improvement will be included in the revised paper. With these updates, EconGym can now scale to support dynamic interactions of up to **100,000 individuals** within a tolerable sampling time. The original test table is shown in Appendix A.2, Table 7.
>
>
>
> **Table: Step time (ms)  at varying population sizes (N).**
>
> | Environment                | N=10  | N=100 | N=1,000 | N=10,000 | N=100,000 | N=1,000,000 |
> | -------------------------- | ----- | ----- | ------- | -------- | --------- | ----------- |
> | delayed_retirement         | 0.415 | 0.743 | 1.284   | 7.434    | 38.021    | 428.524     |
> | personal_pension           | 0.390 | 0.782 | 1.207   | 4.841    | 35.478    | 398.457     |
> | population_aging           | 0.389 | 0.727 | 1.193   | 4.838    | 41.252    | 412.033     |
> | pension_gap                | 0.438 | 0.899 | 1.606   | 7.641    | 66.533    | 692.047     |
> | pension_across_countries   | 0.389 | 0.746 | 1.121   | 4.809    | 40.473    | 400.084     |
> | consumption_tax            | 0.307 | 0.297 | 0.538   | 2.369    | 21.308    | 223.003     |
> | estate_tax                 | 0.406 | 0.743 | 1.073   | 4.793    | 37.928    | 457.300     |
> | universal_basic_income     | 0.384 | 0.752 | 1.207   | 4.846    | 38.486    | 441.864     |
> | optimal_tax                | 0.238 | 0.256 | 0.473   | 2.182    | 21.137    | 241.364     |
> | wealth_tax                 | 0.309 | 0.256 | 0.534   | 2.619    | 21.253    | 215.794     |
> | negative_interest          | 0.245 | 0.246 | 0.422   | 2.362    | 21.090    | 220.007     |
> | inflation_control          | 0.553 | 0.570 | 0.770   | 2.633    | 21.360    | 219.598     |
> | QE                         | 0.658 | 0.566 | 0.780   | 2.833    | 20.592    | 216.561     |
> | optimal_monetary           | 0.626 | 0.573 | 0.765   | 2.776    | 21.396    | 216.744     |
> | dbl_government             | 0.284 | 0.262 | 0.458   | 2.191    | 20.885    | 212.618     |
> | technology                 | 0.513 | 0.790 | 1.119   | 4.612    | 37.385    | 378.126     |
> | monopoly                   | 0.593 | 0.670 | 0.848   | 2.580    | 21.840    | 218.916     |
> | oligopoly                  | 0.628 | 0.581 | 0.985   | 4.069    | 25.975    | 282.529     |
> | monopolistic_competition   | 0.729 | 0.621 | 1.096   | 5.044    | 38.837    | 405.933     |
> | work_hard                  | 0.391 | 0.754 | 1.136   | 5.433    | 36.568    | 413.765     |
> | age_consumption            | 0.773 | 1.857 | 1.418   | 5.658    | 38.316    | 411.893     |
> | asset_allocation           | 0.643 | 0.648 | 0.753   | 2.992    | 22.213    | 229.047     |
> | work_life_well_being       | 0.469 | 0.907 | 1.068   | 5.081    | 42.909    | 404.245     |
> | over_leveraged_consumption | 0.809 | 1.061 | 1.394   | 5.595    | 50.099    | 414.194     |
> | market_type                | 0.255 | 0.297 | 0.426   | 2.595    | 21.794    | 230.198     |
>
>
>
> ---
>
>
>
> > **W2: "I am curious whether and how this platform could be used for incentive design, for example at the level of designing agent rewards or other incentive mechanisms."**
>
> **Answer:**
>
> We sincerely appreciate your insightful question. We believe **EconGym provides an experimental platform for studying *incentive design* in economic contexts**, with agents optimizing self-interested objectives. For example:
>
> - **Government agents** can adjust fiscal, monetary, or pension policies to guide large numbers of individuals, firms, and banks toward goals such as welfare improvement, income equality, or long-term sustainability. Tasks related to macroeconomic policy, such as `delayed_retirement`, `optimal_tax`, and `optimal_monetary`, can be used to experiment with these mechanisms.
> - **Firms** can dynamically set wages and prices, making it possible to study how different pricing mechanisms affect consumer behavior (`market_type`) or how wage setting influences labor supply (`work_hard`, `work_life_well_being`).
>
> EconGym **offers** researchers a rich set of economic modeling environments for studying incentive design, enabling the observation of induced equilibrium behavior in a dynamic, multi-agent economy. Researchers **can also modify reward functions** to dynamically adjust their research questions. The current reward function, designed based on economic theory, is detailed in Appendix C.
>
>
>
> ---
>
> **We hope these clarifications and new results address your concerns. If you have any further questions, we are happy to discuss them.**

---

> > ### Comment · Reviewer_WNuG · 2025-08-04
> >
> > Thank the authors for addressing my questions. I have no further questions. Again, I believe this work has great impact for AI in Econ. I will maintain my score.

---

> > > ### Author Response · Authors · 2025-08-04
> > > **Thank You for Your Positive and Supportive Feedback**
> > >
> > > **Dear Reviewer WNuG**,
> > >
> > > Thank you for your recognition and support of our work. We appreciate your insightful questions and are pleased that our responses have addressed your concerns. The additional experiments and discussions will be incorporated into the revised paper.
> > >
> > > Once again, thank you for your encouraging and supportive feedback.

---

### Official Review · Reviewer_6hA4 · 2025-06-07

**Rating:** 5
**Confidence:** 3

**Summary:**

The paper introduces EconGym, a scalable modular AI testbed addressing limitations of existing economic simulation platforms (e.g., narrow task scope, lack of cross-domain capabilities). Grounded in rigorous economic theory, EconGym integrates 11 heterogeneous role types (households, firms, banks, governments) and enables flexible composition of over 25 economic tasks. Key contributions include: 1) a unified theoretical framework for micro-macro dynamics; 2) modular design supporting cross-domain policy coordination (e.g., fiscal-monetary-pension integration); 3) support for diverse algorithms (RL, LLM, hybrid) and scalability to 10k agents. Experiments show EconGym excels in pension optimization and policy coordination, with hybrid AI-economic strategies outperforming pure learning models. It provides a rigorous testbed for AI-driven economic policy research, balancing realism and efficiency to enable systematic benchmarking of AI algorithms in complex economic scenarios.

**Dataset Code Accessibility:**

Partly

**Ethical Considerations:**

No, there are no or only very minor ethics concerns

**Limitations Weaknesses:**

Part 1: Technical and Code Feedback
I attempted to run the environment code provided in the appendix. Overall, the user experience was suboptimal. While I recognize the project's ambitious scope and the significant engineering effort required, which may explain some of the current issues, substantial improvements are needed in the following areas:
1. Specific Code Quality Issues (Exemplified using agents/ppo_agent file):
  - Critical sections of the code lack necessary comments. For instance, the class definition around Line 54 and its numerous parameters lack clear explanations of their function and purpose. This significantly hampers the understanding and usability of the environment.
  - Around Line 133, there is a substantial amount of commented-out code that is unused and unexplained. This reduces code readability and increases maintenance overhead. It should be either removed entirely or, if intentionally retained for a specific reason, explicitly commented to explain its status and purpose.
  - A significant portion of the code contains comments in Chinese. It is strongly recommended to convert all comments (and future development documentation) to English.
2. Critical Documentation Gap:
  - The environment urgently requires a comprehensive, user-friendly API Reference or Quick Start Guide. While the existing documentation (e.g., document/Fiscal Policy Issues/Q1-Can-consumption-taxes-boost-growth-and-fairness.md) provides valuable background and result demonstrations, it entirely lacks crucial practical guidance on how to use the framework:
    - Reproducing Experiments: How does one configure environment parameters to precisely replicate the experimental results presented in the paper? What are the specific steps for setting up and running these experiments?
    - Modifying the Environment: How can existing environment rules or parameter settings be altered or adjusted?
    - Extending the Environment: How should new Agent types or rules be added? What are the API interface specifications for designing new Agents or rules? How can these APIs be correctly utilized for integration?
  - The absence of this core documentation severely limits the environment's reusability, extensibility, and efficiency of collaboration among researchers. When compared to mature frameworks in the reinforcement learning domain (e.g., OpenSpiel), the current environment exhibits a significant gap in terms of developer tooling maturity and the completeness of user support materials (especially API documentation and tutorials). Sustained development effort and updates are essential to bridge this gap.

---
Part 2: Experimental and Methodological Feedback (Focus on LLM Agent Analysis)
Lack of In-depth Analysis of LLM Agent Decision-Making: A key contribution of the paper lies in the integration of LLM-driven Agents into economic simulations. For traditional RL agents, whose internal decision-making is typically a black box, solely presenting final converged results is an acceptable practice. However, a crucial and distinctive advantage of LLM Agents is their inherent capability to generate interpretable, logically rich textual reasoning during the decision-making process (e.g., Chain-of-Thought, Self-Reflection). This process data holds significant value as it could be leveraged to:
- Understand Causation: Delve into the underlying reasons and logical chains behind agent decisions within specific economic contexts.
- Investigate Behavioral Influences: Explore how agent economic behavior is shaped and modulated by System Prompt settings (e.g., background knowledge, behavioral style, risk preferences).
- Uncover Dynamic Patterns: Analyze how agent decision patterns and interactions evolve under varying spatio-temporal conditions or rule configurations, revealing potential underlying economic dynamics.
- It is regrettable that the experimental section of the paper and its supplementary materials contain no presentation, discussion, or systematic experimental analysis of the LLM Agents' decision-making processes (i.e., textual outputs or reasoning steps). This omission leads to:
  - An inability to verify the rationality of the LLM Agents' decision logic or to probe the intrinsic mechanisms of their behavior.
  - A missed opportunity to leverage the unique interpretability strengths of LLMs to provide deeper insights and significantly strengthen the research claims.
Consequently, this neglect of the distinctive analytical dimension offered by LLM Agents somewhat weakens the depth of the presented research value and the perceived innovativeness of the contribution.
Suggestions:
- It is strongly recommended to include and analyze representative textual outputs (or statistical summaries thereof) from the LLM Agents' decision-making processes in the revised manuscript or supplementary materials.
- Consider designing and reporting experiments that utilize System Prompts to control specific agent characteristics (e.g., risk attitudes, social preferences) and analyze the impact of these settings on simulation outcomes and agent behavior.
- Discuss or analyze variations in agent decision patterns across different economic states, exploiting the interpretability of LLMs to deepen the understanding of the research problem.

**Strengths Contributions:**

The paper's key strengths and contributions lie in the innovative EconGym platform, which overcomes limitations of existing tools (e.g., AI Economist, TaxAI) by enabling flexible composition of 25+ economic tasks through modular design of 11 heterogeneous roles (households, firms, etc.). It uniquely integrates rigorous economic theories (OLG model, Ramsey framework) with AI algorithms (RL, LLM, hybrids), supporting scalable simulations up to 10k agents. Experiments validate its superiority in cross-domain policy coordination (e.g., fiscal-monetary-pension integration) and pension optimization, where hybrid AI-economic strategies outperform pure learning approaches.

The work clearly differentiates itself by addressing prior platforms' narrow scope and lack of cross-domain capabilities (Table 1), demonstrating enhanced realism and efficiency through detailed experiments (e.g., GDP dynamics, Wasserstein Distance metrics). The well-structured presentation, supported by informative figures/tables, highlights its role as a standardized testbed for AI-driven economic research, fostering interdisciplinary advances at the intersection of AI and economics.

---

> ### Author Rebuttal · Authors · 2025-07-30
>
> **Dear Reviewer 6hA4,**
>
> Thank you for your thoughtful review and valuable feedback. We greatly appreciate your recognition of EconGym as an "innovative platform," highlighting its "enhanced realism and efficiency," "well-structured presentation," and its potential for "fostering interdisciplinary advances at the intersection of AI and economics."
>
> In response to your comments, we have made the following improvements and additions:
>
> - Updated the **Python code** and **documentation** based on your suggestions and by learning from established frameworks.
> - **Included representative examples of LLM-generated content** to showcase the decision-making process and reasoning capabilities.
>
> **Below, we provide detailed responses to each of your concerns, and we remain committed to continually improving EconGym to enhance its interpretability and usability as an AI testbed for economic tasks.**
>
> ---
>
>
>
> > **W1: "Part 1: Technical and Code Feedback"**
>
> **Answer:**
>
> Thank you for your detailed and constructive feedback!
>
> **1. Code Optimization:**
>  We have **added** the necessary English comments, **removed** commented-out code, and **made** several other improvements to enhance the readability and usability of EconGym.
>
> **2. Documentation Optimization:**
>  Although the `EconGym/README.md` includes a Quick Start Guide and instructions for modifying parameter files, we have further improved the documentation for each task by:
>
> - **Providing** detailed descriptions of task-specific parameters
> - **Including** instructions on how to modify these parameters
> - **Adding** clear steps for reproducing the experimental results presented in the paper
> - **Providing** guidelines for setting up new agent algorithms, including a framework for custom agent Python files
>
> Due to the rebuttal's constraints, we cannot update the anonymous GitHub repository during this period. However, we fully acknowledge and appreciate your suggestions. We are committed to continuous development and will prioritize improvements to EconGym's usability, reusability, and extensibility.
>
> ---
>
> > **W2: "Part 2: Experimental and Methodological Feedback (Focus on LLM Agent Analysis)"**
>
> Thank you for your thoughtful feedback on our paper. We greatly appreciate your emphasis on the inherent strengths of LLM-driven agents, particularly their capability to generate interpretable, logically rich textual reasoning during the decision-making process, as seen in Chain-of-Thought (CoT) and Self-Reflection.
>
> In response, **we have included a series of representative examples generated by the LLM (`"DeepSeek-V3-0324"`) as a pension authority in the aging-pension scenario, incorporating Chain-of-Thought (CoT) reasoning to illustrate the LLM's decision-making process.** These examples aim to address your concerns and will be fully incorporated into the revised paper.
>
> **1. Understanding Causation: Explaining Policy Adjustments**
>
> The LLM justifies its policy decisions by clearly explaining the cause-and-effect relationships behind each adjustment. For example, when addressing the need for pension system sustainability, the LLM ties the increase in retirement age and contribution rate to specific economic indicators:
>
> > "To address this, a modest increase in the retirement age (from 62.0 to 62.5) is proposed to slightly extend working years without overly burdening workers, thereby sustaining the pension system and supporting GDP growth."
>
> This highlights the LLM’s ability to reason through economic conditions and explain its decisions accordingly.
>
> **2. Adapting to System Prompts: Sensitivity to Economic Conditions**
>
> The LLM demonstrates a keen sensitivity to system prompts such as inflation and GDP growth. When faced with high inflation, the LLM adjusts the contribution rate while ensuring that disposable income is not significantly impacted:
>
> > "The contribution rate is increased from 0.15 to 0.16 to improve pension fund sustainability, as the current fund ratio is 0.0, indicating underfunding. This adjustment is cautious to avoid significantly reducing disposable income and consumption, given the high inflation (18.104%)."
>
> This shows how the LLM tailors its policy decisions to accommodate economic pressures, balancing long-term sustainability with immediate impacts.
>
> **3. Uncovering Dynamic Patterns: Evolving Decisions Over Time**
>
> The LLM adjusts its decisions as the economic environment evolves. For example, with **GDP growth** at 2.064%, the LLM incrementally raises the retirement age to balance pension sustainability while accommodating a growing labor force:
>
> > "The retirement age is increased from 62.5 to 63.0 to address the high pension burden (21.6% of GDP) and sustain the pension system without overly burdening workers, given the relatively high GDP growth rate (2.064%) and mean household income (155,405.89)."
>
> This demonstrates how the LLM accounts for economic changes and revises policies in response to shifting conditions, emphasizing its ability to adjust dynamically.
>
> **4. Addressing Wealth Inequality: Strategic Adjustments to Consumption and Pension**
>
> The LLM also incorporates **wealth inequality** considerations into its decisions. In response to a high Gini index, the LLM adjusts both the retirement age and contribution rate to reduce inequality while maintaining economic stability:
>
> > "The retirement age is increased slightly from 63.0 to 64.0 to help sustain the pension system without overly burdening workers, given the high wealth Gini index (0.730) and projected pension burden (-0.156). This adjustment balances the need to maintain GDP growth (0.179) while minimizing work burden."
>
> This case illustrates how the LLM incorporates **redistributive policy goals**, making it a powerful tool for addressing socio-economic issues within the simulation.
>
> **5. Long-Term Sustainability: Dynamic Adjustments Over Time**
>
> The LLM carefully balances **short-term needs** with **long-term sustainability goals**. In response to a projected pension burden, it gradually increases the retirement age while also moderating the contribution rate to ensure the pension system remains viable without causing undue burden:
>
> > "The decision to increase the retirement age to 67.0 (from 66.5) is based on the need to sustain the pension system given the high wealth Gini index (0.638) and projected pension burden (-0.452). A modest increase in retirement age helps balance the work burden while ensuring pension sustainability."
>
> This demonstrates the LLM’s **long-term perspective**, allowing for gradual adjustments that align with both **economic stability** and **equity goals**.
>
> ---
>
> **We hope these clarifications and new results address your concerns. If you have any further questions, we are happy to discuss them.**

---

> > ### Comment · Reviewer_6hA4 · 2025-08-01
> >
> > Thank you for your detailed response. The improvements made have largely addressed the concerns I raised earlier. Based on the current feedback, I am inclined to maintain my original rating.

---

> > > ### Author Response · Authors · 2025-08-02
> > > **Sincere thanks for your recognition and support of our work.**
> > >
> > > Thank you very much for your support and recognition of our work! Your feedback has greatly motivated us to further refine and extend this research.

---

### Official Review · Reviewer_AcRB · 2025-07-06

**Rating:** 5
**Confidence:** 4

**Summary:**

This paper introduces a new simulation environment EconGym for benchmarking ML/RL research in economics applications. In contrast with prior simulators which tend to focus on narrowly scoped tasks (mostly tax policy design), EconGym includes the ability to choose from different agent types at both individual and organization levels and simulate different kinds of inter-agent interaction scenarios. In the current implementation, authors have also included the ability to simulate LLM, RL and rule-based agents. Experiments comparing agent's behavior across single-domain and cross-domain scenarios demonstrate the potential for learning based approaches in economics applications.

**Dataset Code Accessibility:**

Yes

**Dataset Code Comments:**

The provided code repository includes a thorough README file explaining usability of the code and the different experiment scenarios that can be simulated. I have also checked some of the source code for the agents and the environment, and it appears that code and data accompanying the submission are readily accessible and reproducible and available in an executable format.

**Ethical Considerations:**

No, there are no or only very minor ethics concerns

**Final Justification:**

I will maintain my initial score and review of this submission. Authors have addressed the concerns raised by me and have also included additional baselines for evaluation of this benchmark. I would support accepting this paper to the conference.

**Limitations Weaknesses:**

1. In this paper, experiments evaluating the performance of ML vs rule-based agents focus primarily on single-agent settings in the sense that these experiments do not include multiple RL / LLM agents simultaneously learning to interact in a multi-player game setting. The authors do not comment on the feasibility or possible challenges of such analysis in EconGym.


2. Although the authors have touched upon the benefits of AI-Economics strategies and the trade-off between realism and modeling efficiency, EconGym is essentially limited by the ability to validate these simulation studies in practice. Therefore it would be valuable to include an evaluation suite for testing the OOD generalization or cross-domain transfer of learning based agent policies.

3. With the addition of more real world datasets to this benchmark (as the author's note in future work), it would also be interesting to study the effect of temporally extended actions, long horizon planning and decision making under partial observability through the lens of multi-agent interactions in EconGym.

**Strengths Contributions:**

**Novelty**: EconGym extends over prior work in economics simulation platforms and provides a scalable (gym-like) interface to use with ML algorithms. It expands the types of scenarios that can be simulated, the different types of agents / entities and their interactions as well as the suite of decision making algorithms that can be deployed.

**Relevance**: EconGym is geared towards a practical application setting for RL algorithms in economics. A realistic simulator for macro and microeconomic event dynamics can be quite valuable in testing out proposed economics policies, validating claims of policy outcomes, etc. and while EconGym has its limitations, as the authors acknowledge, it provides an initial step in that direction.

**Presentation**: The paper is well written, with thorough explanation of the motivation behind EconGym and its current implementation. The results comparing the performance of standard ML/RL/LLM agents in EconGym are contextualized in terms of the specific entities involved, their interaction dynamics and associated challenges to highlight the different aspects of the scenario being tested.

---

> ### Author Rebuttal · Authors · 2025-07-30
>
> **Dear Reviewer AcRB,**
>
> Thank you for your thorough review and valuable feedback. We appreciate your recognition of our work as "well written" and "quite valuable."
>
> In this rebuttal, we have addressed your comments by adding experiments and clarifications to further enhance **EconGym**:
>
> - **Added experiments on MARL and LLM (CoT)** to demonstrate the feasibility of multi-agent interactions in EconGym.
> - **Validated EconGym's results** by comparing them with **established economic patterns, real-world data, and established economic modeling benchmarks**.
> - **Showcased EconGym's suitability** for studying temporally extended actions, long-horizon planning, and decision-making under partial observability in a multi-agent context.
> - **Highlighted plans** to incorporate an evaluation suite for testing the generalization of agent policies as part of future work.
>
> **Below, we provide detailed responses to each of your questions, with the utmost care, to fully address your concerns and ensure the clarity and integrity of our work.**
>
> ---
>
>
>
> > **W1: "In this paper, experiments evaluating the performance of ML vs rule-based agents focus primarily on single-agent settings in the sense that these experiments do not include multiple RL / LLM agents simultaneously learning to interact in a multi-player game setting. The authors do not comment on the feasibility or possible challenges of such analysis in EconGym."**
>
> **Answer:**
>
> Thank you for your insightful feedback on the multi-agent setting.
>
> 1. **Multi-Agent Environment in EconGym**:
>    EconGym is designed as a multi-agent testbed. In Section 4.1 ("aging-pension" scenario), we simulate interactions between 1,000 individuals and a pension authority. Section 4.2 ("multi-government coordination") involves multiple government agents interacting with 1,000 individuals. **These are inherently multi-player tasks** within EconGym. Government agents use independent DDPG or independent PPO, both subtypes of MARL algorithms.
> 2. **Possible challenges**:
>    As shown in Section 4.2, applying classic AI algorithms like IDDPG and IPPO to high-dimensional economic tasks, such as multi-government coordination, is challenging. Such complex problems **may require specialized data-driven algorithms**, potentially informed by economic insights.
> 3. **Feasibility**:
>    To further demonstrate the feasibility of "multiple RL / LLM agents" in EconGym, **we added experiments of MADDPG, LLM and an economic method** in the *optimal tax* task, involving a government agent and 10 individuals.While multi-LLM learning methods are still evolving, we tested LLMs using CoT to analyze observable information and aid decision-making, referred to as LLM (CoT).
>
> | Algorithm                                  | Average Individual Utility | GDP                   | Income Gini | Wealth Gini |
> | ------------------------------------------ | -------------------------- | --------------------- | ----------- | ----------- |
> | Utility-Production Model (Economic method) | **271.42±4.16**            | 5.78e+07±5.58e+06     | 0.43±0.02   | 0.35±0.02   |
> | MADDPG                                     | 153.58±2.33                | 6.58e+07±6.19e+06     | 0.44±0.01   | 0.42±0.02   |
> | LLM (CoT)                                  | 201.33±9.68                | **1.01e+08±2.12e+07** | 0.45±0.02   | 0.44±0.02   |
>
> ---
>
>
>
> > **W2: "Although the authors have touched upon the benefits of AI-Economics strategies and the trade-off between realism and modeling efficiency, EconGym is essentially limited by the ability to validate these simulation studies in practice. Therefore it would be valuable to include an evaluation suite for testing the OOD generalization or cross-domain transfer of learning based agent policies."**
>
> **Answer:**
>
> Thank you for your valuable feedback. We will highlight the **existing validation** between EconGym and real-world data, and consider adding an **evaluation suite** for OOD generalization and cross-domain transfer as a future enhancement.
>
> **1. Comparison with Established Economic Patterns and Real-World Data:**
>
> As noted in Section 4.3 and Fig. 6, we already provide comprehensive validation of our simulation results by comparing the **individual-level distributions of consumption, labor, income, and wealth** against real-world data. These comparisons are standard practices in economic benchmarking:
>
> - **Comparison with Established Economic Patterns:** Fig. 6 (subplots 1 and 2) presents the classic "hump-shaped" consumption pattern and the "inverted U-shaped" labor supply curve, both of which are well-established economic patterns in the literature [1, 2].
> - **Comparison with Real-World Data:** Fig. 6 (subplots 3 and 4) illustrates the income and wealth distributions, with the corresponding Lorenz curves aligning closely with real-world data (black line).
>
> **2. Validation Against Established Economic Modeling Benchmarks:**
>
> We also validate EconGym against the **Aiyagari (1994)** model [3], widely recognized as the foundation of heterogeneous-agent macroeconomic analysis [4]. Using its standard parameters (capital share $\alpha = 0.36$, depreciation $\delta = 0.06$, productivity $Z = 1$), we run the *optimal tax* scenario and compare key outcomes:
>
> | Model          | Income Gini | Wealth Gini |
> | -------------- | ----------- | ----------- |
> | Aiyagari model | 0.30–0.40   | ≈0.80       |
> | EconGym (ours) | 0.314±0.081 | 0.728±0.010 |
>
> **EconGym** replicates the inequality and macroeconomic indicators of this benchmark, aligning with established macroeconomic theory beyond platform-specific settings.
>
> **3.** We agree that incorporating an **evaluation suite for testing the OOD generalization and cross-domain transfer of agent policies** is highly valuable. We consider this as a key direction for enhancing **EconGym’s** capabilities in future developments.
>
> [1] Gourinchas, P.O. (2002). Consumption over the life cycle. *Econometrica*.
>
> [2] MaCurdy, T.E. (1981). Empirical model of labor supply in a life-cycle setting. *J. Political Econ*.
>
> [3] Aiyagari, S.R. (1994). Uninsured risk and aggregate saving. *QJE*.
>
> [4] Heathcote, J., Storesletten, K., & Violante, G.L. (2009). Quantitative macroeconomics with heterogeneous households. *JEL*.
>
>
>
> ---
>
>
>
> > **W3: "With the addition of more real world datasets to this benchmark (as the author's note in future work), it would also be interesting to study the effect of temporally extended actions, long horizon planning and decision making under partial observability through the lens of multi-agent interactions in EconGym."**
>
> **Answer:**
>
> Thank you for your insightful suggestion. **EconGym** provides a testbed for:
>
> 1. **Long-horizon planning**: For example, government economic policies consider long-term growth or social welfare, while individuals balance saving and consumption decisions, driven by long-term utility optimization.
> 2. **Decision-making under partial observability**: Agents in **EconGym** possess private information, such as an individual’s wealth, which is not observable by others.
> 3. **Temporally extended actions**: These are also supported, such as the delayed impact of policy decisions, where their effects on economic growth or inflation fully materialize only after several years.
>
> We agree that these topics are highly interesting and plan to conduct further research in these areas using **EconGym** in the future.
>
> ---
>
> **We hope these clarifications and new results address your concerns. If you have any further questions, we are happy to discuss them.**

---

> > ### Comment · Reviewer_AcRB · 2025-08-06
> >
> > Thank you to the authors for the response and for the additional results. I have no further questions and will maintain my score at the moment.

---

> > > ### Author Response · Authors · 2025-08-06
> > > **Sincere thanks for your recognition and support of our work.**
> > >
> > > **Dear Reviewer AcRB,**
> > >
> > > Thank you very much for recognizing the value of our rebuttal. We’re glad that our response addressed your concerns. We will incorporate all additional results and analyses into the revised paper to ensure greater clarity and rigor.

---

### Official Review · Reviewer_3sJJ · 2025-07-08

**Rating:** 4
**Confidence:** 4

**Summary:**

EconGym presents a modular AI testbed for economic decision-making that implements 11 heterogeneous role types (individuals, governments, banks, firms) across 25+ economic tasks spanning fiscal policy, monetary policy, pension systems, and market competition. The platform enables flexible composition of economic scenarios and supports multiple AI algorithms (RL, LLMs, rule-based, economic methods) with scalability up to 10,000 agents. Key experimental findings include RL agents excelling in long-term optimization tasks, hybrid AI+economics approaches outperforming pure AI methods, and successful demonstration of cross-domain policy coordination revealing both synergies and conflicts between different government authorities. I think that the paper is ambitious and despite the limitations will be a valuable contribution to the economic research community once these limitations are adequately addressed.

**Dataset Code Accessibility:**

Yes

**Dataset Code Comments:**

Yes - The submission provides code access through the anonymous repository.

**Ethical Comments:**

EconGym is a simulation platform that does not involve human subjects, personal data collection, or direct policy implementation. The platform uses publicly available aggregate economic data and demographic statistics without privacy implications. While the research could eventually inform economic policy decisions, the current work represents basic research infrastructure development rather than direct policy recommendations. The authors appropriately acknowledge limitations and do not overclaim about real-world policy applications.

**Ethical Considerations:**

No, there are no or only very minor ethics concerns

**Limitations Weaknesses:**

1. The experimental evaluation lacks fundamental statistical rigor expected for policy-relevant research. No confidence intervals, significance tests, or power analyses are provided, and the reported results appear to be based on single runs or insufficient replications (Section 4). For claims about algorithmic superiority, proper statistical validation with multiple seeds and appropriate corrections for multiple comparisons is essential.

2. The evaluation primarily compares algorithms within the EconGym framework without systematic validation against established economic modeling benchmarks (DSGE models, validated ABM frameworks) or direct comparison with peer platforms like TaxAI or AI Economist using common scenarios. This limits the ability to assess whether claimed performance advantages reflect genuine improvements or platform-specific artifacts.

3. The experimental protocols lack sufficient detail for reproducibility (Section 4). Missing information about hyperparameter selection, initialization procedures, and sensitivity analysis prevents independent validation. The aging-pension scenario (Section 4.1) and multi-government coordination (Section 4.2) experiments would benefit from more rigorous experimental design with pre-registered hypotheses and systematic parameter exploration.

4. While the platform incorporates calibrated individual behaviors using empirical data (2022 SCF), the validation of emergent macroeconomic dynamics against real-world economic outcomes is insufficient. The claims about "high realism" (Section 4.3) lack quantitative benchmarks against established economic indicators or historical policy outcomes.

**Strengths Contributions:**

1. EconGym represents a significant advancement in economic simulation platforms by implementing 11 distinct agent types grounded in rigorous economic theory (Ramsey models, OLG models, various market structures). This substantially exceeds existing platforms like AI Economist (2-4 types) and provides a unified framework for diverse economic scenarios that has been lacking in the field.

2. The platform's ability to model coordinated multi-government scenarios (fiscal + monetary + pension policies) addresses a critical gap in existing economic AI research, which typically focuses on isolated policy domains. The demonstration of policy synergies and conflicts represents valuable insights for both AI and economics communities.

3. Supporting multiple AI paradigms (RL, LLMs, hybrid approaches) within a single framework enables systematic algorithmic comparison that is essential for benchmarking progress in economic AI. The demonstrated scalability to 10,000 agents with maintained efficiency addresses practical deployment considerations.

---

> ### Author Rebuttal · Authors · 2025-07-30
>
> **Dear Reviewer 3sJJ,**
>
> Thank you for your thorough review and insightful feedback. We appreciate your recognition of EconGym's contributions: "significant advancement", "ambitious" goals, and providing "valuable insights."
>
> During the rebuttal, we have made every effort to address your comments through additional experiments, comparisons, and expanded analysis:
>
> - **Added statistical rigor to our experimental evaluation**, incorporating mean ± std, confidence intervals, and significance tests to ensure the reliability of our results.
> - **Added experiments using LLM (deepseek)**, demonstrating that its stronger reasoning capabilities led to improved performance in economic decision-making.
> - **Included comparisons with established economic modeling benchmarks and peer platforms**.
> - **Provided hyperparameter files and a user manual for reproducibility**, both available in the anonymous GitHub repository.
> - **Validated simulated individual distributions** by comparing our simulated results with well-established economic patterns and real-world data.
>
> **Below, we provide detailed responses to each of your questions, with the utmost sincerity, to address every concern.**
>
> ---
>
>
>
> > **W1: "The experimental evaluation lacks fundamental statistical rigor expected for policy-relevant research..."**
>
> **Answer:**
>
> We sincerely appreciate your valuable feedback. To address your concerns, we have made several improvements to the experimental evaluation in Section 4.1 the *aging-pension* scenario:
>
> **1. Statistical Validation**: **We added key statistical metrics, including mean ± standard deviation, confidence intervals and significance tests** for all critical economic indicators. These results are based on five random seeds, ensuring statistical robustness and replicability.
>
> **2. Additional Experiments**: **We conducted an additional experiment** using LLM (deepseek). The deepseek version (`DeepSeek-V3-0324`) exhibits stronger reasoning capabilities than the previously tested LLM (`GPT-3.5-Turbo`), leading to significantly improved performance in economic decision-making. Due to the word limit for the rebuttal, we will include more experimental data and statistical metrics in the revised paper.
>
> | Method                | Total GDP (mean ± std) | Total GDP 95% CI        | Social Welfare (mean ± std) | Social Welfare 95% CI  | Wealth Gini (mean ± std) | Wealth Gini 95% CI |
> | - | - | - | - | - | - | - |
> | Real-data (RA=67)     | 4.059e+10 ± 7.251e+08  | [3.857e+10, 4.260e+10]  | 1.139e+06 ± 5.372           | [1.123e+06, 1.153e+06] | 0.373 ± 0.008            | [0.350, 0.397]     |
> | DDPG                  | 5.513e+10 ± 6.173e+09  | [3.799e+10, 7.227e+10]  | 1.351e+06 ± 8.369           | [1.118e+06, 1.583e+06] | 0.373 ± 0.029            | [0.293, 0.453]     |
> | PPO                   | 4.687e+10 ± 9.095e+08  | [4.434e+10, 4.939e+10]  | 1.300e+06 ± 4.830           | [1.287e+06, 1.313e+06] | 0.345 ± 0.008            | [0.324, 0.366]     |
> | LLM (deepseek)        | 6.742e+10 ± 1.472e+09  | [6.091e+10, 7.376e+10]  | 1.540e+06 ± 6.850           | [1.510e+06, 1.569e+06] | 0.326 ± 0.005            | [0.306, 0.346]     |
> | LLM (GPT-3.5-Turbo)   | 5.289e+10 ± 4.943e+09  | [-9.906e+09, 1.157e+11] | 1.428e+06 ± 5.907           | [6.778e+05, 2.177e+06] | 0.336 ± 0.004            | [0.281, 0.392]     |
> | Rule-based (IMF rule) | 5.621e+10 ± 1.300e+09  | [5.260e+10, 5.983e+10]  | 1.430e+06 ± 7.671           | [1.409e+06, 1.451e+06] | 0.329 ± 0.006            | [0.314, 0.344]     |
>
> (The *Total GDP* and *Social Welfare* refer to the sum of each step over the entire episode, as the sustainability of the algorithm is also an important evaluation metric.)
>
> We present the comparisons of the algorithms with **statistical significance** (p<0.05) from the above algorithms (by Total GDP). The full version will be included in the revised paper.
>
> | Algorithm1     | Algorithm2     | t-stat  | p-value | Significant |
> | - | - | - | - | - |
> | Real-data      | LLM (deepseek) | -16.356 | 0.0005  | *           |
> | Real-data      | PPO            | -5.400  | 0.0008  | *           |
> | Real-data      | Rule-based     | -10.496 | 0.00003 | *           |
> | LLM (deepseek) | PPO            | 11.881  | 0.0005  | *           |
> | LLM (deepseek) | Rule-based     | 5.707   | 0.003   | *           |
> | PPO            | Rule-based     | -5.889  | 0.0006  | *           |
>
> ---
>
>
>
> > **W2: "The evaluation primarily compares algorithms within the EconGym framework without systematic validation against established economic modeling benchmarks or direct comparison with peer platforms."**
>
> **Answer:**
>
> **1. Established economic modeling benchmarks:**
>
> We validate EconGym against the most classical benchmark in the DSGE family: the Aiyagari (1994) model [1], widely recognized as the foundation of heterogeneous-agent macroeconomic analysis [2]. Using its standard parameters (capital share $\alpha = 0.36$, depreciation $\delta = 0.06$, productivity $Z = 1$), we run the optimal tax scenario in EconGym.
>
> | Model          | Income Gini | Wealth Gini |
> | - | - | - |
> | Aiyagari model | 0.30–0.40   | ≈0.80       |
> | EconGym (ours) | 0.314±0.081 | 0.728±0.010 |
>
> EconGym reproduces the inequality and macro indicators of this widely accepted benchmark, confirming alignment with established macroeconomic theory beyond platform-specific settings.
>
> [1] Aiyagari, S. R. (1994). *Uninsured idiosyncratic risk and aggregate saving*. QJE.
>
> [2] Heathcote, Storesletten, Violante (2009). *Quantitative Macroeconomics with Heterogeneous Households*. JEL.
>
> **2. Comparison with peer platforms (AI Economist, TaxAI):**
>
> **As shown in Fig.1**, EconGym surpasses peer platforms in task diversity, scalability, and algorithm support.
> Peer platforms differ in modeling assumptions, scenarios, and evaluation metrics, making absolute numerical results not directly comparable. Following standard benchmarks (e.g., OpenAI Gym), we instead compare **environment sampling efficiency**—a key factor for algorithmic research.
>
> | Platform| Agents | Sample time (ms/step) |
> | - | - | - |
> | AI Economist | 4| 4.03 |
> | TaxAI|100| 2.073|
> | **EconGym**| 100| **0.665**|
>
>
> EconGym achieves significantly faster simulation while supporting larger agent populations and richer scenarios, enabling large-scale policy learning and benchmarking.
>
> ---
>
>
>
> > **W3: "The experimental protocols lack sufficient detail for reproducibility."**
>
> **Answer:**
>
> We sincerely appreciate the reviewer’s valuable feedback.
>
> 1. Reproducibility, we would like to clarify the following:
>
>    - **We provided the detailed configuration files for all tasks**, including environment parameters and hyperparameters, publicly available on the anonymous GitHub repository under the `EconGym/cfg/*` directory. These files allow users to run the respective tasks directly. The detailed instructions for reproducing the code are provided in the `EconGym/README.md`.
>
>    - **We also provided a user manual for the 25 economic tasks** in the `EconGym/document/*` directory, including recommended roles, agent algorithms, experimental designs, and baseline results.
>
> 2. For the aging-pension scenario, we added more details:
>
>    - **Initialization:** The private state of all individuals are initialized by sampling from the SCF 2022 dataset, using an OLG model. We rely on U.S. birth rates (2022) and CDC’s stepwise mortality rates (2022) for this initialization.
>    - **Experiment Design:**
>      - 1000 individuals, using behavior cloning policies trained on SCF 2022 data.
>      - The government, acting as a pension authority, sets the retirement age. We compared baselines:
>        -  Real-world retirement ages (60, 63, 65, 67, 70)
>        -  LLM-driven agent
>        -  RL (DDPG algorithm)-dirven agent
>        -  The rule-based agent used the IMF  adjustment rule, an economic method based on government debt.
>    - **Sensitivity Analysis:** By testing different retirement ages, Fig. 3 showed delaying retirement (e.g., retirement age = 70) leads to GDP growth and a longer pension fund lifespan, but at the cost of reduced social welfare.
>
> Due to rebuttal character limits, more experimental details, including the multi-government coordination scenario, will be included in the revised paper.
>
> ---
>
>
>
> > **W4: "While the platform incorporates calibrated individual behaviors using empirical data (2022 SCF), the validation of emergent macroeconomic dynamics against real-world economic outcomes is insufficient."**
>
> **Answer:**
> **In Section 4.3 and Fig. 6, we compared the individual-level distributions of consumption, labor, income, and wealth against real-world data**. This type of comparison is a common approach for validating individual-level distributions in economic benchmarking. Specifically, we provided the following validation:
>
> **1. Comparison with Established Economic Patterns:** Fig. 6 (subplots 1 and 2) showcases the classic "hump-shaped" consumption pattern and the "inverted U-shaped" labor supply curve, which are **well-established economic patterns** in the literature [3,4].
>
> **2. Comparison with Real-World Data:** Fig. 6 (subplots 3 and 4) demonstrates the income and wealth distributions, with the corresponding Lorenz curves **matching the real-world data** (black line).
>
> These comparisons provide strong evidence for the realism of individual-level dynamics within our model. Additionally, as mentioned in our response to Q2, the comparison with **established economic modeling benchmarks** further supports this point.
>
>
>
> [3] Gourinchas, P.O. (2002). Consumption over the life cycle. *Econometrica*.
>
> [4] MaCurdy, T.E. (1981). Empirical model of labor supply in a life-cycle setting. *J. Political Econ*.
>
>
>
> ---
>
> **We hope these clarifications and new results will encourage you to reconsider your score. If you have any further questions, we are happy to discuss them.**

---

> > ### Comment · Reviewer_3sJJ · 2025-08-04
> >
> > Thank you for the detailed and thoughtful rebuttal. I appreciate the authors’ thorough efforts to address the concerns I raised through additional experiments, improved statistical rigor, expanded comparisons, and enhanced reproducibility. I have no further questions and look forward to seeing the final version of the paper.

---

> > > ### Author Response · Authors · 2025-08-05
> > > **Thank You for Acknowledging Our Efforts in Addressing Your Concerns**
> > >
> > > **Dear Reviewer 3sJJ**,
> > >
> > > We are deeply grateful for your thoughtful engagement with our work and for confirming that our rebuttal addressed your concerns. We are very glad that the additional experiments, statistical analyses, and expanded comparisons were helpful.
> > >
> > > We will **incorporate** all of these additional results and analyses into the revised paper to ensure the final version is as clear and rigorous as possible.
> > >
> > > If it would be appropriate, may we kindly ask whether you might consider **raising your score** now that your concerns have been addressed? If there are any further points you believe we should include to strengthen the revision, please let us know, and we will promptly integrate them.
> > >
> > > With best regards!

---

### Decision · Program_Chairs · 2025-09-18

**Decision:**

Accept (poster)

**Comment:**

This paper introduces EconGym, a scalable, modular testbed for AI-driven economic policy research that instantiates 11 heterogeneous roles and composes them into 25+ tasks. All reviewers recommend accepting the paper. They all agree that this paper proposes an unified “role + task” design which enables composing complex, multi-government scenarios rather than isolated, single-policy settings. Supporting for RL, LLMs, rule-based, and hybrid methods allows apples-to-apples comparisons, and the evaluation is thorough. After carefully reading the paper, rebuttal, and reviews, the AC agrees with the reviewers on accepting the paper.

===== FINAL UPDATE FROM DB Track PCs ====

The final decision for this paper has been taken by the program chairs after consultation with the SACs. All Senior Area Chairs have ranked papers according to the feedback from the AC during the review process. We decided to leave the original meta-review to reflect the opinion of the AC in light of the initial discussions with reviewers and SAC.